# Detecting and Measuring Confounding Using Causal Mechanism Shifts

**Abbavaram Gowtham Reddy**
Indian Institute of Technology Hyderabad
cs19resch11002@iith.ac.in

**Vineeth N Balasubramanian**
Indian Institute of Technology Hyderabad
vineethnb@iith.ac.in

## Abstract

Detecting and measuring confounding effects from data is a key challenge in causal inference. Existing methods frequently assume causal sufficiency, disregarding the presence of unobserved confounding variables. Causal sufficiency is both unrealistic and empirically untestable. Additionally, existing methods make strong parametric assumptions about the underlying causal generative process to guarantee the identifiability of confounding variables. Relaxing the causal sufficiency and parametric assumptions and leveraging recent advancements in causal discovery and confounding analysis with non-i.i.d. data, we propose a comprehensive approach for detecting and measuring confounding. We consider various definitions of confounding and introduce tailored methodologies to achieve three objectives: (i) detecting and measuring confounding among a set of variables, (ii) separating observed and unobserved confounding effects, and (iii) understanding the relative strengths of confounding bias between different sets of variables. We present useful properties of a confounding measure and present measures that satisfy those properties. Our empirical results support the usefulness of the proposed measures.

## 1 Introduction

Understanding the underlying causal generative process of a set of variables is crucial in many scientific studies for applications in treatment and policy designs [44]. While randomized controlled trials (RCTs) and causal inference through active interventions are ideal choices for understanding the underlying causal model [19, 12, 13, 55], RCTs and/or active interventions are often impossible/infeasible, and some times unethical [50, 6]. Research efforts in causal inference hence rely on observational data to study causal relationships [44, 59, 65, 18, 41]. However, recovering the underlying causal model purely from observational data is challenging without further assumptions; this challenge is further exacerbated in the presence of unmeasured confounding variables.

A confounding variable is a variable that *causes* two other variables, resulting in a spurious association between those two variables. As exemplified with *Simpson's paradox* [58] and many other studies [20, 1, 31], the presence of confounding variables is an important quantitative explanation for why *correlation does not imply causation*. It is challenging to observe and measure all confounding variables in a scientific study [60, 44]. Identifying *latent* or *unobserved* confounding variables is even more challenging, and misinterpretation presents various challenges in downstream applications, such as discovering causal structures from observational data. Numerous methods operate under the assumption of *causal sufficiency* [45, 4, 60, 8, 51, 65], implying the non-existence of unobserved confounding variables. Causal sufficiency presupposes that all pertinent variables required for causal inference have been observed. However, this may not be a practical or testable assumption.

The study of confounding has various applications, chief among them being causal discovery - identifying the causal relationships among variables [38, 40, 63]. It is also useful for determining whether a set of observed confounding variables is sufficient to adjust for estimating causal effects [29], measuring the extent to which statistical correlation between variables can be attributed to confound-

ing [24, 25, 62], and verifying the comparability of treatment and control groups in non-randomized interventional studies [16].

A fundamental problem in causal inference tasks lies in detecting hidden confounding variables from observational data alone. However, this is non-trivial and poses various challenges. For example, a key issue is that given a marginal distribution over observed variables, there are infinitely many joint distributions corresponding to causal graphs involving unobserved variables [56]. To tackle such challenges, recent endeavors show that using data from different environments helps in improved causal discovery [40, 38, 33, 45, 23], detecting causal mechanism shifts [36], and detecting unobserved confounding [29, 38]. However, such recent efforts often subsume confounding detection under causal discovery, focusing primarily on identifying confounding factors while overlooking other useful information, such as the relative strength of confounding between variable sets and the distinction between observed and unobserved confounding within a variable set. We seek to address these gaps in this work.

We focus exclusively on the problem of studying confounding from multiple perspectives, including (i) detecting and measuring confounding among a set of variables, (ii) assessing the relative strengths of confounding among different sets of observed variables, and (iii) distinguishing between observed and unobserved confounding among a set of variables. *The primary focus of causal inference often lies in verifying the presence or absence of confounding rather than determining the exact value of the measured confounding. However, we leverage the measured confounding to assess the relative strengths of confounding between sets of variables. To achieve the above objectives, we utilize data from various contexts, where each context results from shifts in the causal mechanisms of a set of variables [38, 45].* This allows us to propose different measures of confounding based on the available context information. Our contributions can be summarized as follows.

- For various definitions of confounding, we propose corresponding measures of confounding and present useful properties of the proposed measures. To our knowledge, this is the first comprehensive study that examines various aspects of observed and unobserved confounding using data from multiple contexts without making parametric or causal sufficiency assumptions.
- We study pair-wise confounding, confounding among multiple variables, how to separate unobserved confounding from overall confounding, and present ways to assess relative confounding.
- We present an algorithm for detecting and measuring confounding using data from multiple contexts. Experimental results are performed to verify theoretical analysis.

## 2   Related Work

The study of confounding has typically been embedded as part of causal discovery algorithms in most existing work. Causal discovery methods can be categorized according to several criteria, including the type of data utilized (observational versus interventional/experimental), parametric versus non-parametric approaches, or whether they relax causal sufficiency assumptions [65, 59]. Considering our focus in this work on studying confounding comprehensively by going beyond observed confounding variables, we discuss literature that are directed towards methods that relax the causal sufficiency assumption and rely on experimental data.

**Causal Discovery via Observational Data, Relaxing Causal Sufficiency:** Constraint based causal discovery algorithms produce equivalence class of graphs that satisfy a set of conditional independence constraints [60, 11, 9, 42]. Other methods such as [2, 28, 27] reduce the problem complexity by assuming a parametric form of the underlying causal model (e.g., variables are jointly Gaussian in Chandrasekaran et al. [7]), thereby returning unique causal graphs. Nested Markov Models (NMMs) [56, 57, 49, 14] allow identifiability of causal models with latent factors by using (pairwise) Verma constraints. A recent approach using differentiable causal discovery [2] combines NMMs with the differentiable constraint [66] to discover a partially directed causal network and likely confounded nodes. Unlike these methods, our focus in this work is on detecting and measuring confounding under various settings, instead of recovering the entire causal graph or equivalence class.

**Causal Discovery Using Data From Multiple Environments:** Given access to a set of observed confounding variables, very recent work [29] presented testable conditional independence tests that are violated only when there is unobserved confounding. However, their analysis is focused towards the downstream causal effect estimation. We aim to provide a unified framework for studying and measuring confounding under different types of contextual information available.

Other methods [33, 23] learn an equivalence class of graphs when data from observational and interventional distribution are available. Confounding has also shown to be detected in linear models with non-Gaussian variables [20]. In linear models, a spectral analysis method was proposed in [25] to understand to what extent the statistical correlation between a set of variables on a target variable can be attributed to confounding. See Tab. 4 of [40] for an overview of causal discovery methods that use data from multiple environments or contexts. Under the specific assumptions of causal sufficiency and sparse mechanism shift, a method was proposed in [45] to reduce the size of a given Markov equivalence class using mechanism shift score. A differentiable causal discovery method was proposed in [4] to use interventional data to recover interventional Markov equivalence class. While these methods use data from different contexts, they assume the absence of unobserved confounding variables; we instead focus on capturing both observed and unobserved confounding.

**Measuring and Interpreting Confounding:** Earlier efforts in the field have studied different measures for observed confounding, each tailored to address specific challenges [44, 15, 35, 3, 39, 30, 43, 34]. Such measures have also been refined to address specific issues [24, 54]; for e.g., a method to correct the non-linearity effect present in confounding estimates via the exposure–outcome association with and without adjustment for confounding was proposed in [24]. In contrast, we measure the effects of both observed and *unobserved* confounding. Motivated from the *ignorability* property in potential outcomes framework [61, 26], the divergence between nominal and complete propensity density has been considered as an indicative of hidden confounding [26]. To the best of our knowledge, the efforts closest to ours are [38, 40], which study confounding using data from multiple contexts without the causal sufficiency assumption. However, they *do not measure confounding* and detect confounding only as a step to discover the causal graph. Ours is a more general framework for studying and measuring confounding from multiple perspectives.

In regression models, certain difference thresold between the coefficients of treatment variable before and after adjusting for the possible confounding is considered as the indication for the presence of confounding. This process of choosing a threshold is also called *change-in-estimate* criterion. Typical threshold used in literature is $10\%$ [54, 32, 5].

## 3  Background and Problem Setup

Let $\mathbf{X}$ be a set of observed variables and $\mathbf{Z}$ be a set of unobserved or latent variables. The values of $\mathbf{X}, \mathbf{Z}$ can be real, discrete, or mixed. Let $\mathcal{G}$ be the underlying directed acyclic graph (DAG) among the variables $\mathbf{V} = \mathbf{X} \cup \mathbf{Z}$. Directed edges among the variables in $\mathbf{V}$ indicate direct causal influences. Assume that the set of unobserved variables $\mathbf{Z}$ are jointly independent and are exogenous to $\mathbf{X}$ (i.e., $Z_i \perp\!\!\!\perp Z_j$ and $X_k \not\rightarrow Z_j \ \forall i, j, k$). In this setting, any two nodes $X_i, X_j \in \mathbf{X}$ sharing a common parent $Z_k \in \mathbf{Z}$ are said to be confounded, and $Z_k$ is said to be a confounding variable. For a node $X_i \in \mathbf{X}, \mathbf{PA}_i = \{X_j \in \mathbf{X} | X_j \rightarrow X_i\} \cup \{Z_j \in \mathbf{Z} | Z_j \rightarrow X_i\}$ denotes the set of parents of $X_i$.

For a node $X_i$, $\mathbb{P}(X_i | \mathbf{PA}_i)$ is called the *causal mechanism* of $X_i$. The causal mechanism $\mathbb{P}(X_i | \mathbf{PA}_i)$ encodes how the variable $X_i$ is influenced by its parents $\mathbf{PA}_i$. Following earlier work [22, 38, 45, 21, 46, 52], we make the following general assumption about the underlying causal mechanisms of data.

**Assumption 3.1.** *(Independent Causal Mechanisms [44, 47]) A change in $\mathbb{P}(X_i | \mathbf{PA}_i)$ has no effect on and provides no information about $\mathbb{P}(X_j | \mathbf{PA}_j) \ \forall j \neq i$.*

Identifying confounding from only observational data is challenging without further assumptions [28]. Hence, following earlier work [38, 29, 40], we assume that the data over the variables $\mathbf{X}$ is observed over multiple *contexts or environments*. While there are various ways of formulating/constructing contexts, in this paper, we assume that each context is created as a result of either *hard (a.k.a. structural)* interventions or *soft (a.k.a. parametric)* interventions on a subset $\mathbf{V}_S \subseteq \mathbf{V}$ of variables where $S$ is a set of indices. Performing hard intervention on a variable $V_i$ is the same as setting the value of $V_i$ to a value $v_i$. Hard intervention on a variable $V_i$ removes the influence of its parents $\mathbf{PA}_i$ on $V_i$. Performing soft intervention on a variable $V_i$ is the same as changing the causal mechanism of $V_i$, $\mathbb{P}(V_i | \mathbf{PA}_i)$, with a new causal mechanism $\tilde{\mathbb{P}}(V_i | \mathbf{PA}_i)$. Soft intervention on a variable $V_i$ does not remove the influence of its parents $\mathbf{PA}_i$ on $V_i$. The idea of explicitly considering context information and using different contexts as context variables to create extended causal graphs has been studied in the literature. Context variables are also called as *policy variables, decision variables, regime variables, domain variables, environment variables, etc.* [40, 45, 17, 22].

Let $\mathbf{C} = \{c_1, c_2, \ldots, c_n\}$ be the set of $n$ contexts and let $\mathbb{P}^c(\mathbf{X}), c \in \mathbf{C}$, denotes the probability distribution of the observed variables $\mathbf{X}$ in the context $c$. Let $\mathbf{C}_{S \wedge R}$, where $S, R$ are sets of indices, be

the set of contexts in which we observe mechanism changes for the set of variables $\mathbf{X}_{S \cup R}$. Similarly, let $\mathbf{C}_{S \wedge \neg R}$ be the set of contexts in which we observe mechanism changes for the set of variables $\mathbf{X}_S$ but not for the variables $\mathbf{X}_R$. We say that the causal mechanism of a variable $X_i$ changes between two contexts $c, c'$ if $\mathbb{P}^c(X_i|\mathbf{PA}_i) \neq \mathbb{P}^{c'}(X_i|\mathbf{PA}_i)$. Given the data over observed variables in each context, there exist methods for detecting mechanism shifts of each variable between the contexts [36, 38, 45, 37]. For example, the $p$-value$(\mathbb{P}^c(X_i|\mathbf{PA}_i^o) \neq \mathbb{P}^{c'}(X_i|\mathbf{PA}_i^o))$ where $\mathbf{PA}_i^o$ is the set of observed parents of $X_i$ can be used to detect mechanism change for $X_i$ between the contexts $c, c'$ [38, 36]. Hence, we focus on detecting and measuring confounding among a set of variables, assuming that the causal mechanism shifts are observed among that set of variables.

Context information is not very useful if there is no restriction on how causal mechanisms are changed between the contexts [45, 38]. For example, the causal mechanisms of $X_i$ and $X_j$ both differing across all (or no) contexts would trivially satisfy Assumption 3.1, but reveal no information about the underlying causal mechanisms [10, 38]. Hence, following earlier work [45, 38, 17], we make the following assumptions.

**Assumption 3.2.** *(Sparse Causal Mechanism Shift [53]) Causal mechanisms of variables change sparsely across contexts, i.e., if $p := (\mathbb{P}^c(X_i|\mathbf{PA}_i) \neq \mathbb{P}^{c'}(X_i|\mathbf{PA}_i))$, then $0 < p < 0.5; \quad \forall c, c' \in \mathbf{C}$.*

Assumption 3.2 implies that the causal mechanisms change infrequently across contexts. This assumption is more general because, in many scientific studies, for any given context, interventions typically affect only a few variables [53].

**Assumption 3.3.** *(Markov Property under Mechanism Shifts [17]) The distribution $\mathbb{P}(\mathbf{V})$ is given by $\mathbb{P}(\mathbf{V}) = \int \mathbb{P}^C(\mathbf{V})d\mathbb{P}(C) = \int \Pi_i \mathbb{P}^C(V_i|\mathbf{PA}_i)d\mathbb{P}(C)$. In other words, variables $\mathbf{V}$ are assumed to be conditionally exchangeable, so that the same graph $\mathcal{G}$ applies in every context $c \in \mathbf{C}$.*

**Assumption 3.4.** *(Causal Sufficiency Over $\mathbf{X} \cup \mathbf{Z}$) All common parents of any pair of observed nodes belong to the set $\mathbf{X} \cup \mathbf{Z}$. In other words, all relevant variables for detecting confounding and the unobserved confounding variables are already present in $\mathbf{X} \cup \mathbf{Z}$.*

***Problem Statement:*** *Given data over the observed variables $\mathbf{X}$ in multiple contexts, each context resulting from a sparse causal mechanism shift of variables in $\mathbf{V}$, (i) can we identify which pairs or sets of variables are confounded and can we measure the confounding strength? (ii) can we isolate the confounding effects of observed and unobserved confounding variables? and (iii) can we study the relative strengths of confounding among different sets of variables?*

To address the above problem, in the next section, we consider various definitions of confounding and present appropriate confounding measures depending on the context information available.

## 4 Detecting and Measuring Confounding

In this section, we present methods for detecting and measuring confounding for various scenarios in which shifts in causal mechanisms are observed. Considering any three observed variables $X_i, X_j, X_o \in \mathbf{X}$ and an unobserved confounding variable $Z \in \mathbf{Z}$, we present mea-

| Settings | Confounding Definition Based On | Required Context Information | Type of Intervention |
|---|---|---|---|
| 1 | Directed Information [48] & Noncollapsibility [15, 43, 54] | $\mathbf{C}_{\{i\} \wedge \neg P_{ij}}$ $\mathbf{C}_{\{j\} \wedge \neg P_{ji}}$ | Hard / Structural |
| 2 & 3 | Mutual Information | $\mathbf{C}_{\{i\} \wedge \{j\}}$ | Soft / Parametric |

Table 1: Summary of the various settings for detecting and measuring confounding between $X_i, X_j$. Here $P_{ij}$ is the set of node indices that belong to a path from $X_i$ to $X_j$ including $j$.

sures of confounding depending on the information about mechanism shifts of $X_i, X_j, X_o, Z$. Each of the following subsections includes: (i) a definition of confounding, (ii) a corresponding definition of the confounding measure, (iii) a method for isolating the unobserved confounding measure from the overall confounding, (iv) an extension of the confounding measure to more than two variables, and (v) key properties of the proposed confounding measures. See Tab. 1 and Fig. 1 for an overview.

### 4.1 Setting 1: Measuring Confounding Using Directed Information Between $X_i, X_j$.

In this setting, we use the fact that directed information does not vanish in the presence of a confounding variable [64, 48]. To this end, we leverage the interventional effects of $X_i, X_j$ on each other to define a measure of confounding.

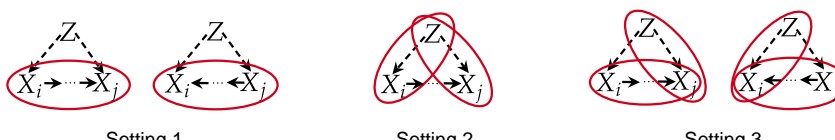

Figure 1: **Setting 1:** When contexts $\mathbf{C}_{\{i\} \wedge \neg P_{ij}}$ and $\mathbf{C}_{\{j\} \wedge \neg P_{ji}}$ are known where $P_{ij}$ is the set of node indices that belong to a path from $X_i$ to $X_j$ including $j$, we leverage directed information from $X_i$ to $X_j$ and from $X_j$ to $X_i$ to define a measure of confounding (Defn. 4.4). **Setting 2:** Causal mechanism changes in $Z$ introduces dependencies on the observed distributions of $X_i, X_j$. We leverage such dependencies to measure confounding when contexts $\mathbf{C}_{\{i\} \wedge \{j\}}$ are known (Defn. 4.6). **Setting 3:** If we know that there is a causal path from $X_i$ to $X_j$, we leverage dependencies between the pairs $(X_i, X_j)$ and $(Z, X_j)$ to measure confounding. Similarly, if we know that there is a causal path from $X_j$ to $X_i$, we leverage dependencies between the pairs $(X_i, X_j)$ and $(Z, X_i)$ to measure confounding (Defn. 4.7). Dashed arrows from $Z$ indicate that $Z$ is unobserved.

**Definition 4.1.** *(Directed Information [48]). The directed information $I(X_i \to X_j)$ from $X_i \in \mathbf{X}$ to $X_j \in \mathbf{X}$ is defined as the conditional Kullback-Leibler divergence between the distributions $\mathbb{P}(X_i|X_j), \mathbb{P}(X_i|do(X_j))$. That is:*

$$I(X_i \to X_j) := D_{KL}(\mathbb{P}(X_i|X_j)||\mathbb{P}(X_i|do(X_j))) := \mathbb{E}_{\mathbb{P}(X_i, X_j)} \log \frac{\mathbb{P}(X_i|X_j)}{\mathbb{P}(X_i|do(X_j))} \qquad (1)$$

**Definition 4.2.** *(No Confounding [44]) When measuring the causal effect of a (treatment) variable $X_i$ on a (target) variable $X_j$, the ordered pair $(X_i, X_j)$ is unconfounded if and only if the directed information from $X_j$ to $X_i$: $I(X_j \to X_i)$ is zero. Equivalently, $\mathbb{P}(X_j|X_i) = \mathbb{P}(X_j|do(X_i))$.*

A similar definition of confounding that relates the conditional distribution $\mathbb{P}(X_i|X_j)$ and interventional distribution $\mathbb{P}(X_i|do(X_j))$ is defined as follows.

**Definition 4.3.** *(Noncollapsibility) [15, 43, 54] The statistical association between two variables $X_i$ and $X_j$ is said to be noncollapsible if the association strength differs in each level/strata of other variable $X_k$. That is, if $X_k$ is a confounding variable between $X_i, X_j$, we have $\mathbb{P}(X_j|X_i) \neq \mathbb{P}(X_j|do(X_i)) = \mathbb{E}_{X_k}(\mathbb{P}(X_j|X_i, X_k))$.*

From Defns. 4.1 and 4.2, for a pair of variables $(X_i, X_j)$, observing $I(X_j \to X_i) > 0$ and $I(X_i \to X_j) > 0$ implies that $\mathbb{P}(X_j|do(X_i)) \neq \mathbb{P}(X_j|X_i)$ and hence the presence of confounding (see Tab. 2). Using the above properties of directed information, we measure *confounding* as follows.

**Definition 4.4.** *(Confounding Measure 1) When causal mechanism shifts of two variables $X_i, X_j \in \mathbf{X}$ are observed, resulting in different contexts, under the Assumptions 3.2-3.4, the measure of confounding $CNF\text{-}1(X_i, X_j)$ between $X_i$ and $X_j$ is defined as follows.*

$$CNF\text{-}1(X_i, X_j) := 1 - e^{-\min(I(X_i \to X_j), I(X_j \to X_i))} \qquad (2)$$

For all the confounding measures, we use exponential transformation to limit the range of the measure between $0$ and $1$. Note that in a DAG, one of $I(X_i \to X_j), I(X_j \to X_i)$ is zero under no confounding (see Tab. 2 for a simple example with two and three node graphs). Hence $CNF\text{-}1(X_i, X_j)$ outputs zero when there is no confounding between $X_i, X_j$. Similarly $CNF\text{-}1(X_i, X_j)$ outputs positive real value in the range $(0, 1]$ when there is confounding. We leverage data from multiple contexts to evaluate $\mathbb{P}(X_i|X_j)$ and $\mathbb{P}(X_i|do(X_j))$ as follows. In this

| | **Graph** | $I(X_i \to X_j)$ | $I(X_j \to X_i)$ |
|---|---|---|---|
| Uncnf. | $X_i \to X_j$ | $> 0$ | $= 0$ |
| | $X_j \to X_i$ | $= 0$ | $> 0$ |
| Confounded | $X_i \to X_j$ 
 $Z \to X_i, Z \to X_j$ | $> 0$ | $> 0$ |
| | $X_j \to X_i$ 
 $Z \to X_i, Z \to X_j$ | $> 0$ | $> 0$ |

Table 2: Directed information values in two and three node graphs. If $X_i, X_j$ are confounded by $Z$, we observe positive directed information from both directions.

setting, we assume each context is generated as a result of *hard* interventions on a subset of variables. Let $P_{ij}$ be the set of node indices that belong to a path from $X_i$ to $X_j$ including $j$, we use the contexts $\mathbf{C}_{\{i\} \wedge \neg P_{ij}}$ to evaluate $\mathbb{P}(X_j|do(X_i))$ as $\mathbb{P}(X_j|do(X_i)) = \mathbb{E}_{c \in \mathbf{C}_{\{i\} \wedge \neg P_{ij}}}[\mathbb{P}^c(X_j|X_i)]$. Intuitively, to compute the interventional effects of $X_i$ on $X_j$, we need to observe mechanism changes only for $X_i$ to account for the potential causal influence from $X_i$ to $X_j$. In addition, none of the nodes in a causal path from $X_i$ to $X_j$ should be intervened. We use observational data to evaluate $\mathbb{P}(X_j|X_i)$.

**Proposition 4.1.** *(Identifiability of $\mathbb{P}(X_j|do(X_i))$) $\mathbb{P}(X_j|do(X_i))$ is identifiable from the set of contexts $\mathbf{C}_{\{i\}\wedge\neg P_{ij}}$. To detect and measure confounding between a pair of nodes $X_i, X_j$, it is enough to observe two sets of contexts $\mathbf{C}_{\{i\}\wedge\neg P_{ij}}$ and $\mathbf{C}_{\{j\}\wedge\neg P_{ji}}$. Thus, $n$ sets of contexts are needed to detect and measure confounding between $\binom{n}{2}$ distinct pairs of nodes in a causal DAG with $n$ nodes.*

When a confounding variable $X_o$ between $X_i, X_j$ is observed, and there may exist an unobserved confounding variable $Z$, it is crucial to detect and measure unobserved confounding effect [29]. We utilize conditional directed information to define the measure of unobserved confounding.

**Definition 4.5.** *(Conditional Directed Information [48]). The conditional directed information $I(X_i \rightarrow X_j|X_o)$ from $X_i$ to $X_j$ conditioned on $X_o$ is defined as the conditional Kullback-Leibler divergence between the distributions $\mathbb{P}(X_i|X_j, X_o), \mathbb{P}(X_i|do(X_j), X_o)$ as follows.*

$$I(X_i \rightarrow X_j|X_o) := D_{KL}(\mathbb{P}(X_i|X_j, X_o)||\mathbb{P}(X_i|do(X_j), X_o)) := \mathop{\mathbb{E}}_{\mathbb{P}(X_i,X_j,X_o)} \log \frac{\mathbb{P}(X_i|X_j, X_o)}{\mathbb{P}(X_i|do(X_j), X_o)} \quad (3)$$

This measure can trivially be extended to the case where there exist multiple observed and unobserved confounding variables. The expression $\mathbb{P}(X_i|do(X_j), X_o)$ means conditioning on $X_o$ in the interventional distribution $\mathbb{P}(X_i|do(X_j))$. Now, the conditional confounding can be measured as:

$$CNF\text{-}1(X_i, X_j|X_o) := 1 - e^{-\min(I(X_i \rightarrow X_j|X_o), I(X_j \rightarrow X_i|X_o))} \quad (4)$$

Intuitively, by conditioning on an observed confounding variable $X_o$, we control the association between $X_i, X_j$ flowing via $X_o$ and measure the influence via the unobserved confounding variables.

**Beyond Pairwise Confounding:** We now study when a set $\mathbf{X}_S$ of variables where $|\mathbf{X}_S| > 2$ are jointly confounded i.e., share a common confounding variable and how to measure the joint confounding among the variables $\mathbf{X}_S$.

**Theorem 4.1.** *A set of observed variables $\mathbf{X}_S$ are jointly unconfounded if and only if there exists three variables $X_i, X_j, X_k \in \mathbf{X}_S$ such that $I(X_i \rightarrow X_j|X_k) = I(\{X_i, X_k\} \rightarrow X_j)$.*

We now define the measure of confounding among the variables in $\mathbf{X}_S$ as follows.

$$CNF\text{-}1(\mathbf{X}_S) = \sum_{i \in S} CNF\text{-}1(\mathbf{X}_{S\setminus\{i\}}, X_i) \quad (5)$$

Conditional confounding among a set of variables can be defined similarly to Eqn. 4. We now study some useful properties of the measure $CNF$-1.

**Theorem 4.2.** *For any three observed variables $X_i, X_j, X_o$ and an unobserved confounding variable $Z$, the following statements are true for the measure $CNF$-1.*

1. *(**Reflexivity and Symmetry.**) $CNF\text{-}1(X_i, X_i|X_o) = 0$, $CNF\text{-}1(X_i, X_j|X_o) = CNF\text{-}1(X_j, X_i|X_o)$.*

2. *(**Positivity.**) $CNF\text{-}1(X_i, X_j) > 0$ if and only if $X_i, X_j$ are confounded. Given an observed confounding variable $X_o$ between $X_i, X_j$, $CNF\text{-}1(X_i, X_j|X_o) > 0$ if and only if there exists an unobserved confounding variable $Z$ between $X_i, X_j$.*

3. *(**Monotonicity.**) $CNF\text{-}1(X_i, X_j) > CNF\text{-}1(X_k, X_l)$ implies that the pair of variables $X_i, X_j$ are more strongly confounded than the pair of variables $X_k, X_l$ in the sense of Defns. 4.2 and 4.3.*

## 4.2 Setting 2: Detecting and Measuring Confounding Using the Mechanism Shifts of $Z$.

The previous setting utilizes the interventional effects of $X_i(X_j)$ on $X_j(X_i)$ to define a measure of confounding between $X_i, X_j$. In this setting, we utilize the association between the observed marginal distributions of $X_i, X_j$ under causal mechanism shifts of $Z$ to measure confounding. To this end, similar to [38], we make the following assumption.

**Assumption 4.1.** *(Shift Faithfulness [38]) Let $Z$ be a common parent for a set of variables $\mathbf{X}_S \subseteq \mathbf{X}$. Then each causal mechanism shift in $Z$ between two contexts $c, c'$ entails a causal mechanism change in each $X_i \in \mathbf{X}_S$ between the same contexts $c, c'$.*

One consequence of the Assumption 4.1 is that a change in the causal mechanism of $Z$ induces correlations between the expectations of $X_i, X_j$ in different contexts. To understand this, consider the following structural equations.

$$Z \sim \mathcal{N}(\mu(c), \sigma^2(c)) \qquad X_i := \alpha Z + \epsilon_i \qquad X_j := \beta X_i + \gamma Z + \epsilon_j \quad (6)$$

Where $c$ denotes the context and $\epsilon_x$ and $\epsilon_y$ are noise variables with zero mean and have no additional restriction on the underlying probability distribution. The causal graph corresponding to this model has the nodes $X_i, X_j, Z$ and edges: $Z \to X_i, Z \to X_j, X_i \to X_j$. It is easy to see that $\mathbb{E}(X_i) = \alpha\mu(c)$ and $\mathbb{E}(X_j) = (\alpha\beta + \gamma)\mu(c)$. Following Assumption 4.1, whenever there is a change in causal mechanism of $Z$ (e.g., $c$ changes to $\tilde{c}$ in Eqn. 6), there is a change in both $\mathbb{E}(X_i), \mathbb{E}(X_j)$. Additionally, since $Z$ is a common cause of both $X_i, X_j$, there is a spurious association between $\mathbb{E}(X_i), \mathbb{E}(X_j)$. Subsequently, in the set of contexts $\mathbf{C}_{\{i\}\wedge\{j\}}$ the values $\mathbb{E}(X_i), \mathbb{E}(X_i)$ are spuriously associated. Under Assumptions 3.2 and 4.1, restricting our analysis to $\mathbf{C}_{\{i\}\wedge\{j\}}$ ensures that with high probability, the association between $\mathbb{E}(X_i), \mathbb{E}(X_i)$ is due to the confounding variable $Z$. In this example, the association between $\mathbb{E}(X_i), \mathbb{E}(X_j)$ exists even if $\beta = 0$, i.e., $X_i \not\to X_j$. To define confounding measure, we create two random variables $E_i^C, E_j^C$ which we define as $E_i^C = \mathbb{E}_{X_i \sim \mathbb{P}^c(X_i)}(X_i), E_j^C = \mathbb{E}_{X_j \sim \mathbb{P}^c(X_j)}(X_j)$ respectively where $c \in \mathbf{C}_{\{i\}\wedge\{j\}}$. Relying on the context information $\mathbf{C}_{\{i\}\wedge\{j\}}$ and utilizing the association between $E_i^C$ and $E_j^C$, we define a confounding measure as follows.

**Proposition 4.2.** *(Confounding Based on Mutual Information) If two variables $X_i, X_j$ are confounded by a variable $Z$, the induced random variables $E_i^C, E_j^C$ as described above have non zero mutual information $I(E_i^C; E_j^C)$.*

**Definition 4.6.** *(Confounding Measure 2) When the causal mechanism shifts are observed for $X_i, X_j$ in different contexts and the contexts $\mathbf{C}_{\{i\}\wedge\{j\}}$ are known, under the Assumptions 3.2-4.1, the measure of confounding $CNF\text{-}2(X_i, X_j)$ between $X_i$ and $X_j$ is defined as*

$$CNF\text{-}2(X_i, X_j) := 1 - e^{-I(E_i^C; E_j^C)} \tag{7}$$

To measure the unobserved confounding strength when we already observe a confounding variable $X_o$, we condition on the observed confounding variable $X_o$ to define $CNF\text{-}2(X_i, X_j | X_o)$ as follows.

$$CNF\text{-}2(X_i, X_j | X_o) := 1 - e^{-I(E_i^C; E_j^C | X_o)} \tag{8}$$

**Beyond Pairwise Confounding:** Following earlier work [38], we utilize total correlation among triplets $(E_i^C, E_j^C, E_k^C)$ of random variables in $\{E_i^C\}_{i \in S}$ to verify whether a set of variables $\mathbf{X}_S$ are jointly confounded. By Assumption 4.1, we know that the variables in $\mathbf{X}_S$ jointly confounded only if each pair $X_i, X_j; \ i, j \in S$ is pairwise confounded. If all three variables share the same latent confounding variable $Z$, then knowing about one of $E_i^C, E_j^C, E_k^C$ explains away some of the association between the other two, so that we have $I(E_i^C, E_j^C | E_k^C) < I(E_i^C, E_j^C)$. However, for a triplet $(X_i, X_j, X_k)$, it is possible that, rather than jointly confounded, there may be three disjoint confounding variables $Z_{12}, Z_{13}, Z_{23}$ confounding each of the individual pairs: $(X_i, X_j), (X_j, X_k), (X_k, X_i)$. In general, for a set of variables of size $s$ to permit such an equivalent explanation, we would need to have a total of $\binom{s}{2}$ confounding variables with $s(s-1)$ outgoing edges to obtain the same structure of pairwise confounding [38]. While this may plausibly occur for small sets of variables that appear to be pairwise correlated, we assume the true graph $\mathcal{G}$ to be causally minimal in the following sense.

**Assumption 4.2.** *(Confounder Minimality [38]) For every subset $\mathbf{X}_S$ of at least $|S| \geq 4$ variables, there are at most $2|S|$ edges incoming into $\mathbf{X}_S$ from latent confounding variables with at least three children in $\mathbf{X}_S$.*

Assumption 4.2 ensures that variables that appear to be jointly confounded are indeed confounded. In other words, when a small number of latent variables suffice to explain the observed correlations, there should indeed exist only few confounding variables. With this assumption, we can guarantee that joint confounding can be identified from the total correlation.

**Theorem 4.3.** *Let $\mathbf{X}_S$ be a set of variables such that all $X_i, X_j \in \mathbf{X}_S$ are pairwise confounded. Then $\mathbf{X}_S$ is jointly confounded if and only if for each triple $X_i, X_j, X_k \in \mathbf{X}_S$ we have $I(E_i^C; E_j^C | E_k^C) < I(E_i^C; E_j^C)$.*

Now, the measure of joint confounding among a set of variables $\mathbf{X}_S$ can be defined using total correlation $T(E_i^C, \dots, E_{|S|}^C)$ as follows. To evaluate the following expression, we need to use the contexts $\mathbf{C}_{\{1\}\cup\cdots\cup\{|S|\}}$ to ensure that with high probability, the association among the variables in $\mathbf{X}_S$ is due to the joint confounding variable $Z$.

$$CNF\text{-}2(\mathbf{X}_S) = 1 - e^{-T(E_i^C, \dots, E_{|S|}^C)} \tag{9}$$

**Theorem 4.4.** *For any three observed variables $X_i, X_j, X_o$ and an unobserved confounding variable $Z$, the following statements are true for the measure $CNF$-2.*

1. *(**Reflexivity and Symmetry.**) $CNF$-2$(X_i, X_i | X_o) = 1 - e^{-H(E_i^C | X_o)}$ $\forall i$ where $H(.|.)$ denotes conditional entropy and $CNF$-2$(X_i, X_j | X_o) = CNF$-2$(X_j, X_i | X_o)$.*

2. *(**Positivity.**) $CNF$-2$(X_i, X_j) > 0$ if and only if $X_i, X_j$ are confounded. Given an observed confounding variable $X_o$ between $X_i, X_j$, $CNF$-2$(X_i, X_j | X_o) > 0$ if and only if there exists an unobserved confounding variable $Z$ between $X_i, X_j$.*

3. *(**Monotonicity.**) $CNF$-2$(X_i, X_j) > CNF$-2$(X_k, X_l)$ implies that the pair of variables $X_i, X_j$ are more strongly confounded than the pair of variables $X_k, X_l$ in the sense of Defn. 4.2.*

### 4.3 Setting 3: Observing the Causal Mechanism Shifts in $Z$ and Known Causal Path Direction Between $X_i$ and $X_j$

Similar to the previous settings, we utilize marginal and conditional distributions of $X_i, X_j$ to define a measure of confounding. By prior knowledge, if we know the direction of causal path between $X_i, X_j$, we can utilize the causal direction to measure confounding as explained below. In addition to the notations $E_i^C, E_j^C$ introduced in the previous setting, let us denote for each $c \in \mathbf{C}_{\{i\} \wedge \{j\}}, \mathbb{E}_{X_i \sim \mathbb{P}^c(X_i | X_j)}(X_i | X_j), \mathbb{E}_{X_j \sim \mathbb{P}^c(X_j | X_i)}(X_j | X_i)$ with $E_{ij}^C, E_{ji}^C$ respectively. We now leverage dependency among these variables to define the measure of confounding. Intuitively, if $X_i \to X_j$ and if we observe a change in the causal mechanisms of both $X_i, X_j$ due to the causal mechanism changes in $Z$, we also observe a change in the causal mechanism $\mathbb{P}(X_j | X_i)$.

**Definition 4.7.** *(**Confounding Measure 3**) When the causal mechanism shifts are observed for $X_i, X_j$ and the causal direction between the nodes $X_i, X_j$ is known, under the Assumptions 3.2-4.1, the measure of confounding $CNF$-3$(X_i, X_j)$ between $X_i \in \mathbf{X}$ and $X_j \in \mathbf{X}$ is defined as*

$$CNF\text{-}3(X_i, X_j) := \begin{cases} 1 - e^{-I(E_{ji}^C; E_j^C)} & if \quad X_i \to \cdots \to X_j \\ 1 - e^{-I(E_{ij}^C; E_i^C)} & if \quad X_j \to \cdots \to X_i \\ CNF\text{-}2(X_i, X_j) & Otherwise \end{cases} \tag{10}$$

To measure the unobserved confounding strength in the presence of an observed confounding variable $X_o$, similar to setting 2, we can modify Eqn. 10 to condition on the variable $X_o$.

**Beyond Pairwise Confounding:** Using the Assumption 4.2, we have the following.

**Theorem 4.5.** *Let $\mathbf{X}_S$ be a set of variables such that all $X_i, X_j \in \mathbf{X}_S$ are pairwise confounded and the causal relationships among each pair $X_i, X_j$. Then $\mathbf{X}_S$ is jointly confounded if and only if for each triple $X_i, X_j, X_k \in \mathbf{X}_S$ we have $I(E_{ij}^C; E_{jk}^C | E_j^C) < I(E_{ij}^C; E_{jk}^C)$.*

Since we have access to random variables $E_{ij}^C$ in addition to $E_i^C, E_j^C$, it is not straightforward to use all of them to measure joint confounding. To keep the measure simple, we let the measure of joint confounding among the variables $\mathbf{X}_S$ be the same as $CNF$-2$(\mathbf{X}_S)$. That is, $CNF$-3$(\mathbf{X}_S) = CNF$-2$(\mathbf{X}_S)$. Setting 3 is an alternative to Setting 2 when we know the direction of the causal path between $X_i, X_j$. Settings 2 and 3 act as complementary to each other in validating the correctness of our analysis.

**Theorem 4.6.** *For any three observed variables $X_i, X_j, X_o$ and an unobserved confounding variable $Z$, the following statements are true for the measure $CNF$-3.*

1. *(**Reflexivity and Symmetry.**) $CNF$-3$(X_i, X_i | X_o) = 1 - e^{-H(E_i^C | X_o)}$ $\forall i$ where $H(.|.)$ denotes conditional entropy and $CNF$-3$(X_i, X_j | X_o) = CNF$-3$(X_j, X_i | X_o)$.*

2. *(**Positivity.**) $CNF$-3$(X_i, X_j) > 0$ if and only if $X_i, X_j$ are confounded. Given an observed confounding variable $X_o$ between $X_i, X_j$, $CNF$-3$(X_i, X_j | X_o) > 0$ if and only if there exists an unobserved confounding variable $Z$ between $X_i, X_j$.*

3. *(**Monotonicity.**) $CNF$-3$(X_i, X_j) > CNF$-3$(X_k, X_l)$ implies that the pair of variables $X_i, X_j$ are more strongly confounded than the pair of variables $X_k, X_l$ in the sense of Defn. 4.2.*

# 5 Algorithm

Algorithm 1 outlines the procedures to measure confounding in all three settings and can be extended to the case where we evaluate conditional confounding and evaluating confounding among multiple variables. We present two real-world examples where our methods can be applied in Appendix § B.

---

**Algorithm 1:** Algorithm for evaluating pairwise $CNF\text{-}1, CNF\text{-}2, CNF\text{-}3$

---

**Data:** Context information $\mathbf{C}_{\{i\}\wedge\neg P_{ij}}, \mathbf{C}_{\{j\}\wedge\neg P_{ji}}, \mathbf{C}_{\{i\}\wedge\{j\}}$, Contextual Datasets $\{\mathcal{D}^c\}_{c\in\mathbf{C}}$.
**Result:** $CNF\text{-}1(X_i, X_j), CNF\text{-}2(X_i, X_j), CNF\text{-}3(X_i, X_j)$
Step 1:  Evaluate $\mathbb{P}(X_i|X_j), \mathbb{P}(X_j|X_i)$ using observational data;
Step 2:  Evaluate $\mathbb{P}(X_i|do(X_j))$ using $\{\mathcal{D}^c\}_{c\in\mathbf{C}_{\{j\}\wedge\neg P_{ji}}}$;
Step 3:  Evaluate $\mathbb{P}(X_j|do(X_i))$ using $\{\mathcal{D}^c\}_{c\in\mathbf{C}_{\{i\}\wedge\neg P_{ij}}}$;
Step 4:  Evaluate $I(X_i \rightarrow X_j), I(X_j \rightarrow X_i)$;
Step 5:  $CNF\text{-}1(X_i, X_j) = 1 - e^{-\min(I(X_i\rightarrow X_j), I(X_j\rightarrow X_i))}$;
Step 6:  Evaluate $E_i^C, E_j^C$ using $\{\mathcal{D}^c\}_{c\in\mathbf{C}_{\{i\}\wedge\{j\}}}$ ;
Step 7:  $CNF\text{-}2(X_i, X_j) = 1 - e^{-I(E_i^c;E_j^c)}$;
Step 8:  Evaluate $E_{ij}^C, E_{ji}^C$ using $\{\mathcal{D}^c\}_{c\in\mathbf{C}_{\{i\}\wedge\{j\}}}$ ;
Step 9:  compute $CNF\text{-}3(X_i, X_j)$ according to Defn. 4.7;
return $CNF\text{-}1(X_i, X_j), CNF\text{-}2(X_i, X_j), CNF\text{-}3(X_i, X_j)$

---

# 6 Experiments and Results

We perform simulation studies to verify the correctness of the proposed measures. All the experiments are run on a CPU. We report the mean and standard deviation of results taken over five random seeds. Code to reproduce the results is presented in the supplementary material. Code is available at https://github.com/gautam0707/CD_CNF.

**Measuring Confounding:** In this set of experiments, we consider the following four causal structures made of three nodes $X_i, X_j, Z$: $\mathcal{G}_1$ : Empty graph over $Z, X_i, X_j$ i.e., nodes are isolated in the graph, $\mathcal{G}_2 : X_i \rightarrow X_j, \mathcal{G}_3 : Z \rightarrow X_i, Z \rightarrow X_j, \mathcal{G}_4 : Z \rightarrow X_i, Z \rightarrow X_j, X_i \rightarrow X_j$. In $\mathcal{G}_1, \mathcal{G}_2$, there is no confounding between $X_i, X_j$ and in $\mathcal{G}_3, \mathcal{G}_4$ there is confounding effect of $Z$ on $X_i$ and $X_j$. Results in Fig. 2 show that our measures output zero when there is no confounding between $X_i, X_j$ and output positive values when $X_i, X_j$ are confounded by a confounding variable $Z$.

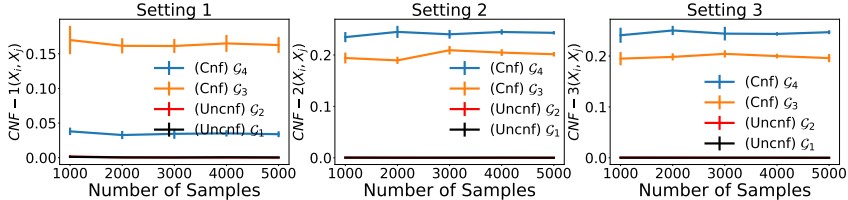

Figure 2: Measure of confounding between a pair of variables $X_i, X_j$. Our measures output zero when there is no confounding between $X_i, X_j$ and output positive values when $X_i, X_j$ are confounded.

**Measuring Conditional Confounding:** We consider the following two causal structures. $\mathcal{G}_5$ : $Z_1 \rightarrow X_i, Z_1 \rightarrow X_j, Z_2 \rightarrow X_i, Z_2 \rightarrow X_j, X_i \rightarrow X_j$. $\mathcal{G}_6$ : $Z \rightarrow X_i, Z \rightarrow X_j, X_i \rightarrow X_j$. In $\mathcal{G}_5$, $X_i$ and $X_j$ are confounded by two variables $Z_1, Z_2$. We measure conditional confounding between $X_i, X_j$ conditioned on $\emptyset, Z_1$, and $Z_2$ respectively. Since confounding still exists in all of the above conditioning settings, $CNF\text{-}2$ correctly returns positive confounding value in all three cases (see Fig. 3 left). On the other hand, in $\mathcal{G}_6$, we measure conditional confounding

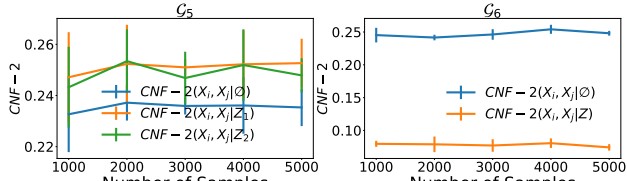

Figure 3: **Left:** Conditioning on one of $\emptyset, Z_1, Z_2$ will not remove confounding between $X_i, X_j$ in $\mathcal{G}_5$. Hence $CNF\text{-}2$ returns positive values. **Right:** In $\mathcal{G}_6$, conditioning on $\emptyset$ does not remove the confounding effect of $Z$ on $X_i, X_j$. Hence, we observe a positive value for $CNF\text{-}2(X_i, X_j|\emptyset)$. Conditioning on $Z$ will block the confounding between $X_i, X_j$. Hence $CNF\text{-}2$ is closer to zero.

between $X_i, X_j$ conditioning on empty set and $Z$. Since conditioning on $Z$ will block the confounding association between $X_i, X_j$, $CNF$-2 returns confounding value closer to zero. However, the unconditioned confounding (conditioning on empty set) value is still large. These results empirically validate the correctness of the proposed measures.

**Downstream Causal Effect Estimation:** For the causal graphs $\mathcal{G}_3, \mathcal{G}_4$, we examine the impact of controlling for nodes identified using our method. We measure the causal effect of $X_i$

| Causal | Not Controlling Confounding | | | | | Controlling Confounding | | | | |
|--------|------|------|------|------|------|------|------|-------|------|-------|
| Graph | 1000 | 2000 | 3000 | 4000 | 5000 | 1000 | 2000 | 3000 | 4000 | 5000 |
| $\mathcal{G}_3$ | 0.55 | 0.57 | 0.55 | 0.52 | 0.52 | 0.06 | 0.02 | 0.007 | 0.03 | 0.009 |
| $\mathcal{G}_4$ | 0.24 | 0.26 | 0.23 | 0.24 | 0.23 | 0.04 | 0.05 | 0.06 | 0.02 | 0.05 |

Table 3: Downstream application of causal effect estimation.

on $X_j$ with and without controlling for the detected confounding variable and report the absolute difference between the true and estimated causal effects in Tab. 3. The results show that controlling for the variables identified by our method reduces the bias in the estimated causal effects.

**Binary Data - Erdös-Rényi Causal Graphs:** To verify the performance of our method on a large scale, similar to [38], we generate causal graphs of various number nodes using Erdös-Rényi model. In these experiments, each context is a result of intervention on one node. This is the reason for having the same value for number of nodes $N$ and number of contexts $|C|$. Sample size denotes the number of data points used in each context. We detect and measure whether each pair of nodes is confounded or not. We then calculate the *Precision, Recall*, and *F1* scores. Our confounding measures obtain good results across all settings.

| | | Setting 1 | | | Setting 2 | | | Setting 3 | | |
|------|-------------|-----------|--------|------|-----------|--------|------|-----------|--------|------|
| $N, |C|$ | Sample Size | Precision | Recall | F1 | Precision | Recall | F1 | Precision | Recall | F1 |
| 10 | 100 | 0.64 | 0.97 | 0.77 | 0.67 | 0.83 | 0.74 | 0.64 | 0.72 | 0.68 |
| 10 | 200 | 0.64 | 1.0 | 0.78 | 0.67 | 0.83 | 0.74 | 0.70 | 0.79 | 0.74 |
| 10 | 300 | 0.64 | 1.0 | 0.78 | 0.67 | 0.83 | 0.74 | 0.65 | 0.76 | 0.70 |
| 10 | 400 | 0.64 | 1.0 | 0.78 | 0.67 | 0.83 | 0.74 | 0.67 | 0.83 | 0.74 |
| 10 | 500 | 0.64 | 1.0 | 0.78 | 0.67 | 0.83 | 0.74 | 0.67 | 0.83 | 0.74 |
| 15 | 100 | 0.81 | 0.95 | 0.88 | 0.80 | 0.85 | 0.82 | 0.80 | 0.79 | 0.80 |
| 15 | 200 | 0.82 | 1.0 | 0.90 | 0.80 | 0.85 | 0.82 | 0.80 | 0.85 | 0.82 |
| 15 | 300 | 0.82 | 1.0 | 0.90 | 0.80 | 0.85 | 0.82 | 0.80 | 0.85 | 0.82 |
| 15 | 400 | 0.82 | 1.0 | 0.90 | 0.80 | 0.85 | 0.82 | 0.80 | 0.85 | 0.82 |
| 15 | 500 | 0.82 | 1.0 | 0.90 | 0.80 | 0.85 | 0.82 | 0.80 | 0.84 | 0.82 |
| 20 | 100 | 0.68 | 0.95 | 0.80 | 0.68 | 0.88 | 0.77 | 0.69 | 0.84 | 0.76 |
| 20 | 200 | 0.69 | 1.0 | 0.82 | 0.68 | 0.88 | 0.77 | 0.68 | 0.87 | 0.76 |
| 20 | 300 | 0.69 | 1.0 | 0.82 | 0.68 | 0.88 | 0.77 | 0.67 | 0.86 | 0.75 |
| 20 | 400 | 0.69 | 1.0 | 0.82 | 0.68 | 0.88 | 0.77 | 0.68 | 0.87 | 0.76 |
| 20 | 500 | 0.69 | 1.0 | 0.82 | 0.68 | 0.88 | 0.77 | 0.68 | 0.87 | 0.76 |
| 25 | 100 | 0.83 | 0.96 | 0.89 | 0.83 | 0.91 | 0.87 | 0.83 | 0.89 | 0.86 |
| 25 | 200 | 0.83 | 1.0 | 0.91 | 0.83 | 0.91 | 0.87 | 0.82 | 0.90 | 0.86 |
| 25 | 300 | 0.83 | 1.0 | 0.91 | 0.83 | 0.91 | 0.87 | 0.83 | 0.91 | 0.87 |
| 25 | 400 | 0.83 | 1.0 | 0.91 | 0.83 | 0.92 | 0.87 | 0.83 | 0.91 | 0.87 |
| 25 | 500 | 0.83 | 1.0 | 0.91 | 0.83 | 0.91 | 0.87 | 0.83 | 0.91 | 0.87 |

Table 4: Results on synthetic datasets for settings 1,2,3.

## 7 Conclusions, Limitations, and Future Work

In this paper, based on the known causal mechanism shifts of observed variables, we propose three measures of confounding along with their conditional and multivariate variants. We also study key properties of these measures. Our measures complement each other depending on the available context information. We propose algorithms to compute the proposed measures and empirically verify their correctness. However, for the same confounded pair of variables, our metrics may yield different results depending on the chosen measure. As discussed in the introduction, the measures are intended to assess the relative strengths of confounding rather than for point-to-point comparison. The number of contexts required to evaluate the measure can be large because many contexts without changes in particular mechanisms are discarded. Identifying appropriate real-world datasets and applying the proposed measures to those datasets is an interesting area for future work, as is developing measures that efficiently use context information. Additionally, devising new definitions for confounding and proposing corresponding confounding measures is also an interesting future direction. We aim to pursue these ideas.

## Acknowledgments

This work was partly supported by the Prime Minister's Research Fellowship (PMRF) program and a Google Research Scholar Award. We are grateful to the anonymous reviewers for their valuable feedback, which improved the presentation of the paper.

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

# Appendix

## A    Proofs

**Proposition 4.1.** *(**Identifiability of** $\mathbb{P}(X_j|do(X_i))$)* $\mathbb{P}(X_j|do(X_i))$ *is identifiable from the set of contexts* $\mathbf{C}_{\{i\}\wedge\neg P_{ij}}$. *To detect and measure confounding between a pair of nodes* $X_i, X_j$, *it is enough to observe two sets of contexts* $\mathbf{C}_{\{i\}\wedge\neg P_{ij}}$ *and* $\mathbf{C}_{\{j\}\wedge\neg P_{ji}}$. *Thus,* $n$ *sets of contexts are needed to detect and measure confounding between* $\binom{n}{2}$ *distinct pairs of nodes in a causal DAG with* $n$ *nodes.*

*Proof.* Since the set of contexts $\mathbf{C}_{\{i\}\wedge P_{ij}}$ consist of data with all possible interventions on $X_i$, if a context $c$ is generated by performing intervention on $X_i$ with the value $x_i$, the expression $\mathbb{P}(X_j|do(X_i = x_i))$ is equal to the expression $\mathbb{P}(X_j|X_i = x_i)$ in that context $c$.

From Defn. 4.4, to detect and measure confounding between the pair of variables $X_i, X_j$, we need to evaluate $\mathbb{P}(X_j|do(X_i))$ and $\mathbb{P}(X_i|do(X_j))$. To this end, from the previous paragraph, we need two sets of contexts $\mathbf{C}_{\{i\}\wedge P_{ij}}$ and $\mathbf{C}_{\{j\}\wedge P_{ji}}$. Following these observations, it is enough to have $n$ sets of contexts to detect and measure confounding between $\binom{n}{2}$ distinct pairs of nodes. $\qquad\square$

**Theorem 4.1.** *A set of observed variables* $\mathbf{X}_S$ *are jointly unconfounded if and only if there exists three variables* $X_i, X_j, X_k \in \mathbf{X}_S$ *such that* $I(X_i \to X_j|X_k) = I(\{X_i, X_k\} \to X_j)$.

*Proof.* Consider three variables $X_i, X_j, X_k$ in the underlying causal graph. Consider the conditional directed information between $X_i, X_j$ given $X_k$ and the subsequent manipulations as follows.

$$
\begin{aligned}
I(X_i \to X_j|X_k) &:= \mathbb{E}_{\mathbb{P}(X_i,X_j,X_k)} \log \frac{\mathbb{P}(X_i|X_j,X_k)}{\mathbb{P}(X_i|do(X_j),X_k)} \\
&= \mathbb{E}_{\mathbb{P}(X_i,X_j,X_k)} \log \left( \frac{\mathbb{P}(X_i,X_k|X_j)}{\mathbb{P}(X_i,X_k|do(X_j))} \times \frac{\mathbb{P}(X_k|do(X_j))}{\mathbb{P}(X_k|X_j)} \right) \\
&= \mathbb{E}_{\mathbb{P}(X_i,X_j,X_k)} \log \frac{\mathbb{P}(X_i,X_k|X_j)}{\mathbb{P}(X_i,X_k|do(X_j))} - \mathbb{E}_{\mathbb{P}(X_j,X_k)} \log \frac{\mathbb{P}(X_k|X_j)}{\mathbb{P}(X_k|do(X_j))} \\
&= I(\{X_iX_k\} \to X_j) - I(X_k \to X_j)
\end{aligned}
$$

Since $I(X_k \to X_j) \geq 0$, we have $I(X_i \to X_j|X_k) \leq I(\{X_iX_k\} \to X_j)$. Equality holds only when $X_k, X_j$ are unconfounded. $\qquad\square$

**Theorem 4.2.** *For any three observed variables* $X_i, X_j, X_o$ *and an unobserved confounding variable* $Z$, *the following statements are true for the measure* $CNF$-1.

1. *(**Reflexivity and Symmetry.**)* $CNF$-1$(X_i, X_i|X_o) = 0$, $CNF$-1$(X_i, X_j|X_o) = CNF$-1$(X_j, X_i|X_o)$.

2. *(**Positivity.**)* $CNF$-1$(X_i, X_j) > 0$ *if and only if* $X_i, X_j$ *are confounded. Given an observed confounding variable* $X_o$ *between* $X_i, X_j$, $CNF$-1$(X_i, X_j|X_o) > 0$ *if and only if there exists an unobserved confounding variable* $Z$ *between* $X_i, X_j$.

3. *(**Monotonicity.**)* $CNF$-1$(X_i, X_j) > CNF$-1$(X_k, X_l)$ *implies that the pair of variables* $X_i, X_j$ *are more strongly confounded than the pair of variables* $X_k, X_l$ *in the sense of Defns. 4.2 and 4.3.*

*Proof.* **Reflexivity:** From the definition of directed information, $I(X_i \to X_i|X_o) = \mathbb{E}_{\mathbb{P}(X_i,X_j,X_o)} log \frac{\mathbb{P}(X_i|X_o)}{\mathbb{P}(X_i|X_o)} = 0$ and hence $CNF$-1$(X_i, X_j|X_o) = 1 - e^0 = 0$.

**Symmetry:** Even if $I(X_i \to X_j|X_o)$ is not symmetric, the expression '$\min(I(X_i \to X_j|X_o), I(X_j \to X_i|X_o))$' is symmetric and hence $CNF$-1$(X_i, X_j|X_o)$ is symmetric.

**Positivity:** If $X_i, X_j$ are confounded, irrespective of the direction of the causal path between $X_i$ and $X_j$, we have $\mathbb{P}(X_i|X_j) \neq \mathbb{P}(X_i|do(X_j))$ and $\mathbb{P}(X_j|X_i) \neq \mathbb{P}(X_j|do(X_i))$. Hence $I(X_i \to X_j) > 0$ and $I(X_j \to X_i) > 0$. We now have $CNF$-1$(X_i, X_j) > 0$. The above statement is true even if there is no causal path between the nodes $X_i, X_j$. The above statements are valid even after conditioning on an observed confounding variable $X_o$ if there is an unobserved confounding between $X_i, X_j$.

**Monotonicity:** Without loss of generality, assume that the inequality $CNF\text{-}1(X_i, X_j) > CNF\text{-}1(X_k, X_l)$ is a result of $I(X_i \to X_j) > I(X_k \to X_l)$. That is, the KL divergence between $\mathbb{P}(X_i|X_j)$ and $\mathbb{P}(X_i|do(X_j))$ is greater than the kl divergence between $\mathbb{P}(X_k|X_l)$ and $\mathbb{P}(X_k|do(X_l))$. That is, the pair of distributions $\mathbb{P}(X_k|X_l)$ and $\mathbb{P}(X_k|do(X_l))$ are closer to each other compared to the pair $\mathbb{P}(X_i|X_j)$ and $\mathbb{P}(X_i|do(X_j))$. As a result, $X_k, X_l$ are closer to being *not confounded* in the sense of Defns. 4.2 and 4.3. $\qquad\square$

**Proposition 4.2. (Confounding Based on Mutual Information)** *If two variables $X_i, X_j$ are confounded by a variable $Z$, the induced random variables $E_i^C, E_j^C$ as described above have non zero mutual information $I(E_i^C; E_j^C)$.*

*Proof.* There are two sources of dependency between $E_i^C, E_j^C$. If $X_i, X_j$ are causally related in the underlying causal model generating the data, there will be a dependency between $E_i^C, E_j^C$ in the context $C_{\{i\} \wedge \{j\}}$ as the interventions are soft. On the other hand, as per the Assumption 4.1, any shift in the causal mechanism of $Z$ leads to a change in both the mechanisms of $X_i, X_j$ leading to a dependency. Hence the random variables $E_i^C, E_j^C$ have non-zero mutual information. $\qquad\square$

**Theorem 4.3.** *Let $\mathbf{X}_S$ be a set of variables such that all $X_i, X_j \in \mathbf{X}_S$ are pairwise confounded. Then $\mathbf{X}_S$ is jointly confounded if and only if for each triple $X_i, X_j, X_k \in \mathbf{X}_S$ we have $I(E_i^C; E_j^C|E_k^C) < I(E_i^C; E_j^C)$.*

*Proof.* Following the Assumption 4.2, when three variables $X_i, X_j, X_k$ are confounded by as single confounding variable $Z$, conditioning on one of $E_i^C, E_j^C, E_k^C$ explains away some of the dependency between other two. Hence we have $I(E_i^C; E_j^C|E_k^C) < I(E_i^C; E_j^C)$ for all triples $i, j, k$. $\qquad\square$

**Theorem 4.4.** *For any three observed variables $X_i, X_j, X_o$ and an unobserved confounding variable $Z$, the following statements are true for the measure $CNF\text{-}2$.*

1. *(**Reflexivity and Symmetry.**) $CNF\text{-}2(X_i, X_i|X_o) = 1 - e^{-H(E_i^C|X_o)} \ \ \forall i$ where $H(.|.)$ denotes conditional entropy and $CNF\text{-}2(X_i, X_j|X_o) = CNF\text{-}2(X_j, X_i|X_o)$.*

2. *(**Positivity.**) $CNF\text{-}2(X_i, X_j) > 0$ if and only if $X_i, X_j$ are confounded. Given an observed confounding variable $X_o$ between $X_i, X_j$, $CNF\text{-}2(X_i, X_j|X_o) > 0$ if and only if there exists an unobserved confounding variable $Z$ between $X_i, X_j$.*

3. *(**Monotonicity.**) $CNF\text{-}2(X_i, X_j) > CNF\text{-}2(X_k, X_l)$ implies that the pair of variables $X_i, X_j$ are more strongly confounded than the pair of variables $X_k, X_l$ in the sense of Defn. 4.2.*

*Proof.* **Reflexivity:** from the definition of mutual information, $I(E_i^C; E_i^C|X_o) = H(E_i^C|X_o) - H(E_i^C|E_i^C, X_o) = H(E_i^C|X_o)$. Substituting in the definition of $CNF\text{-}2(X_i, X_j)$, result follows.

**Symmetry:** The result follows from the 'symmetry' property of mutual information.

**Positivity:** If $X_i, X_j$ are confounded, from the Assumption 4.1, $E_i^C, E_j^C$ are dependent random variables. Hence the mutual information is positive. The result follows after substituting some positive value for $I(E_i^C; E_j^C)$ in the definition of $CNF\text{-}2(X_i, X_j)$. The same argument goes for conditional confounding.

**Monotonicity:** from the definition of $CNF\text{-}2(X_i, X_j)$, $CNF\text{-}2(X_i, X_j) > CNF\text{-}2(X_k, X_l)$ implies $I(E_i^C; E_j^C) > I(E_k^C; E_l^C)$. From the Defn. 4.2, $X_i, X_j$ have higher mutual information than the pair $X_k, X_l$ and hence $X_i, X_j$ are more strongly confounded than $X_k, X_l$. $\qquad\square$

**Theorem 4.5.** *Let $\mathbf{X}_S$ be a set of variables such that all $X_i, X_j \in \mathbf{X}_S$ are pairwise confounded and the causal relationships among each pair $X_i, X_j$. Then $\mathbf{X}_S$ is jointly confounded if and only if for each triple $X_i, X_j, X_k \in \mathbf{X}_S$ we have $I(E_{ij}^C; E_{jk}^C|E_j^C) < I(E_{ij}^C; E_{jk}^C)$.*

*Proof.* Following the Assumption 4.2, when three variables $X_i, X_j, X_k$ are confounded by as single confounding variable $Z$, conditioning on $E_k^C$ explains away some of the dependency between $E_{ij}^C, E_{jk}^C$. Hence we have $I(E_{ij}^C; E_{jk}^C|E_j^C) < I(E_{ij}^C; E_{jk}^C)$ for all triples $i, j, k$. $\qquad\square$

**Theorem 4.6.** *For any three observed variables $X_i, X_j, X_o$ and an unobserved confounding variable $Z$, the following statements are true for the measure $CNF$-3.*

1. *(**Reflexivity and Symmetry.**) $CNF$-3$(X_i, X_i|X_o) = 1 - e^{-H(E_i^C|X_o)}$ $\forall i$ where $H(.|.)$ denotes conditional entropy and $CNF$-3$(X_i, X_j|X_o) = CNF$-3$(X_j, X_i|X_o)$.*

2. *(**Positivity.**) $CNF$-3$(X_i, X_j) > 0$ if and only if $X_i, X_j$ are confounded. Given an observed confounding variable $X_o$ between $X_i, X_j$, $CNF$-3$(X_i, X_j|X_o) > 0$ if and only if there exists an unobserved confounding variable $Z$ between $X_i, X_j$.*

3. *(**Monotonicity.**) $CNF$-3$(X_i, X_j) > CNF$-3$(X_k, X_l)$ implies that the pair of variables $X_i, X_j$ are more strongly confounded than the pair of variables $X_k, X_l$ in the sense of Defn. 4.2.*

*Proof.* **Reflexivity:** from the definition of mutual information, $I(E_{ii}^C; E_i^C|X_o) = I(E_i^C; E_i^C|X_o) = H(E_i^C|X_o) - H(E_i^C|E_i^C, X_o) = H(E_i^C|X_o)$. Substituting in the definition of $CNF$-3$(X_i, X_j)$, result follows.

**Symmetry:** Since we rely on the direction of the causal path between $X_i, X_j$, for a given pair of nodes $X_i, X_j$, we have $CNF$-3$(X_i, X_j) = CNF$-3$(X_j, X_i)$ from Defn. 4.7.

**Positivity:** If $X_i, X_j$ are confounded and $X_i \rightarrow X_j$, from the Assumption 4.1, $E_{ji}^C, E_j^C$ are dependent random variables. Hence the mutual information $E_{ji}^C, E_j^C$ is positive. The result follows after substituting positive value for $I(E_{ji}^C; E_j^C)$ in the definition of $CNF$-3$(X_i, X_j)$. The same argument goes for conditional confounding.

**Monotonicity:** from the definition of $CNF$-3$(X_i, X_j)$, without loss of generality, $CNF$-3$(X_i, X_j) > CNF$-3$(X_k, X_l)$ implies $I(E_{ji}^C; E_j^C) > I(E_{lk}^C; E_l^C)$. From the Defn. 4.2, $X_i, X_j$ have higher mutual information and hence are more strongly confounded than $X_k, X_l$. □

# B   Real-world Examples

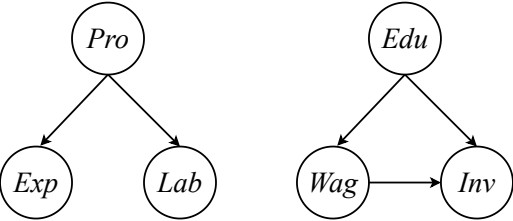

Figure 4: Two real-world examples where our method can be applied. Here *Pro*: Production Volume, *Exp*: Exports, *Lab*: Total Labor Required, *Edu*: Education, *Wag*: Wages, *Inv*: Investments. We can perform interventions on the above variables and any combination thereof to obtain context-specific data. We can use such data to identify and measure confounding by applying our methods.

