# OpenReview forum: "Detecting and Measuring Confounding Using Causal Mechanism Shifts"
_NeurIPS.cc/2024/Conference — NeurIPS 2024 poster_

### Official Review · Reviewer_e4Fq · 2024-07-09

**Soundness:** 2
**Presentation:** 2
**Contribution:** 2
**Rating:** 5
**Confidence:** 4

**Summary:**

This paper introduces some measures of (conditional) confounding, based on information theoretic quantities. Some of their properties and an algorithm to estimate the measures using data from different environments. Although not included in the main text, the authors test their theory on some synthetic data.

**Strengths:**

- The ideas on the paper are original as far as I know and I believe that the question regarding the measurement of confounding is an important one. The authors differentiate their work from previous work.
- Beyond the errors outlined in the weakness section, the paper is well-written and presented. It is a bit unfortunate that the experiments were not included on the main paper.

**Weaknesses:**

- There are several typos/mistakes on the paper:

-- On line 158 and 169: $p$ is defined as the inequality between two distributions, this definition doesn’t make sense. I checked their reference 33 and 35 and it says in there that it is the probability of the distributions being different. Furthermore, p-value is a well-defined and known term, why overloading it?

-- On definition 4.1 what they call directed information is not an -expected- KL divergence, it is simply a KL divergence.

-- On Table 2, on rows 1 and 2 the equality with 0 is inverted. For example, if $X_i\to X_j$, then $P(X_i\mid X_j)=P(X_i\mid do(X_j))$ so that the log of such quantity is 0 for all values of $X_i$ and $X_j$, that is $I(X_i\to X_j)=0$.

-- Is definition 4.6 really a definition? That sounds like a proposition to me.

- I’m unsure of how large is the significance of the contribution being made (see questions and limitations). Which paired with the errors and typos above make me recommend for rejection of the paper.

**Questions:**

I would like to get a feeling for why the properties of their measure of confoundedness are valuable (Theorems 4.2, 4.4 and 4.6). The only really essential property that I can think of such a measure is that it is 0 if the variables are non confounded and not 0 otherwise. Why for example do we want positivity? One could even be interested in a negative measure if the variables have a negative correlation under confoundedness, for example.

**Limitations:**

This research does not contain any obvious negative societal impact. The authors do include a couple of sentences at the end of the paper where they state that there is a challenge to find real-world data to test their theory. Although I value the author's honesty about this limitation, I think it is too strong of a limitation.

---

> ### Author Rebuttal · Authors · 2024-08-07
>
> > On line 158 and 169: p is defined as the inequality between two distributions, this definition doesn’t make sense... why overloading it?
>
> We undestand the concern. To fix this, following [35], we will edit line 158 in the revised manuscript as follows.
>
> *"For example, the $p\text{-value}(\mathbb{P}^c(X_i|\mathbf{PA}_i^o)\neq\mathbb{P}^{c'}(X_i|\mathbf{PA}_i^o))$ where $\mathbf{PA}_i^o$ is the set of..."*
>
> > On definition 4.1 what they call directed information is not an -expected- KL divergence, it is simply a KL divergence.
>
> We agree with you. We will change **"expected KL divergence"** to **"conditional KL-divergence"** in the revised manuscript.
>
> >  On Table 2, on rows 1 and 2 the equality with 0 is inverted...
>
> We beg to differ. If $X_i\rightarrow X_j$, then we have $\mathbb{P}(X_i|X_j) \neq \mathbb{P}(X_i|do(X_j))$ and hence $I(X_i\rightarrow X_j)>0$. This is because, in $X_i\rightarrow X_j$, conditioning on $X_j$ does not make $X_i, X_j$ independent. However, intervening on $X_j$ makes $X_i, X_j$ independent when there is no confounding between $X_i, X_j$. Formally, when there is no confounding, $\mathbb{P}(X_i|do(X_j)) = \mathbb{P}(X_i)$. On the other hand, $\mathbb{P}(X_i|X_j)\neq \mathbb{P}(X_i)$.
>
> > Is definition 4.6 really a definition? That sounds like a proposition to me.
>
> We agree with you, we will change the definition to proposition. The proof is trivial and follows from the definition of mutual information, which we will include in the revised manuscript.
>
> > I would like to get a feeling for why the properties of their measure of confoundedness are valuable...
>
> We believe that these properties are essential to ensure the correctness of the proposed metrics from a measurement perspective. The monotonicity property is crucial for understanding which set of variables are more confounded than others. Since our metrics are derived from positive quantities like directed information and mutual information, they return positive values. Studying negative confounding is a potential future direction.
>
> > The authors do include a couple of sentences at the end of the paper where they state... I think it is too strong of a limitation.
>
> We'd like to point out that we only didn't focus on real-world application, and rather deferred it to future work to apply our methods to real-world datasets. We have included two simple real-world examples in the uploaded rebuttal PDF (Figure 1) demonstrating the applicability of our methods (the purpose of these examples is only to show its practical value, and not intended to be comprehensive or complex). For example, we can study settings 2 and 3 using the SACHS dataset [35,37]. To study setting 1, it is enough to check for datasets that are obtained from randomized control trials where a set of variables are intervened.
>
> We will modify the line in conclusions to reflect this point.
>
> References:
> [35] Mameche, Sarah, Jilles Vreeken, and David Kaltenpoth. "Identifying Confounding from Causal Mechanism Shifts." International Conference on Artificial Intelligence and Statistics. PMLR, 2024.
> [37] Mooij, Joris M., Sara Magliacane, and Tom Claassen. "Joint causal inference from multiple contexts." Journal of machine learning research 21.99 (2020): 1-108.
>
> We hope our responses address your concerns. We will update the manusript accordingly. We are happy to discuss further if you have any additional questions.

---

> > ### Comment · Reviewer_e4Fq · 2024-08-08
> > **Answer to the rebuttal**
> >
> > I thank the authors for taking the time to answer my questions.
> >
> > About Table 2: you are absolutely right, I got confused with the causal graphs and the interventions when I was trying the measures out.
> >
> > About the real world examples and the properties of the definition: I appreciate those, and I would be ok if the paper was purely theoretical and had no experimental results, but then in that case I would expect very strong theoretical results, which in my opinion is not what the authors provide. In fact, the authors' answer to my question about the feeling of the properties seems to be that the properties are post-hoc; that is, they define some measure of confounding using information-theoretic quantities, they realise the properties hold because they defined it in such a way and then they write some Theorems based on that. In other words, I would say the authors have not answered my question of why the properties (besides monotonicity) are important for measuring confounding. I am happy to discuss this further, if the authors are interested in that.

---

> > > ### Author Response · Authors · 2024-08-10
> > > **Response to the reviewer**
> > >
> > > Thank you for your reply. Please note that **we do not** cast our paper as a purely theoretical one. Due to space constraints, we had to move the experimental results to the appendix. The rebuttal PDF also includes  additional experimental results. We will include the experimental results in the main paper, using the additional page provided in the final version if our work gets accepted.
> > >
> > > There are four properties of our measures that we propose: *reflexivity*, *symmetry*, *positivity*, and *monotonicity*. We are glad that you already acknowledged the use of *positivity* and our explanation in the rebuttal for *monotonicity*.
> > >
> > > *Positivity* and *monotonicity* properties are **not post-hoc** results because the measures themselves are motivated by the definitions of confounding (Definitions 4.2, 4.3, 4.6). Based on these definitions, we propose measures whose values can be compared to find the relative strength of confounding between any two sets of variables. If it helps the reader, in the revised manuscript, after the definitions of confounding, we will introduce positivity and monotonicity as important properties that any confounding measure should satisfy and then introduce the definitions of our proposed measures.
> > >
> > > *Reflexivity* and *symmetry* act as validations for the correctness of the proposed measures and cater to extreme cases such as measuring the confounding between a variable and itself (reflexivity) and measuring the confounding by changing the order of variables (symmetry). We would be happy to clarify this, or reduce the emphasis on these two properties in the revised manuscript.
> > >
> > > We are happy to discuss further if you have any additional questions.

---

> > > > ### Comment · Reviewer_e4Fq · 2024-08-13
> > > > **On the properties**
> > > >
> > > > I thank the authors once again for their answer.
> > > >
> > > > So in fact I haven't accepted positivity as a property that I would like to hold in a measurement of confounding. In fact, out of all the properties, this is the one that I disagree the most with (I think your argument on symmetry and reflexivity about avoiding some extreme cases is a valid one) and, as we discussed above monotonicity, is also reasonable.
> > > >
> > > > Since the time for discussion is limited, I will increase my score to 5. But from my point of view, it is worth including a discussion of why these properties are important (at least to you, maybe you disagree with me that having a sign on a confounding measure is valuable).

---

> > > > > ### Author Response · Authors · 2024-08-14
> > > > >
> > > > > Thank you for engaging in the discusion and increasing the score. We would like to clarify our perspective on the positivity property of the confounding measure.
> > > > >
> > > > > We first would like to state that **we do not disagree** with your view that negative values of confounding can be valuable. We clarify that we focus on the **strength of confounding** in our work, which by itself has not been done before. Our perspective to the positivity property comes naturally from the definition of confounding found in literature [14,40,41,51]. Specifically, definitions 4.2 and 4.6 (defn 4.6 will be changed to a proposition as you suggested) lead to positive values of confounding. We simply adhere to existing work and definitions of confounding that quantify the strength of confounding in this paper.
> > > > >
> > > > > We understand that the discussion time window will soon close. In summary, we highlight the following contribution of this paper.
> > > > >
> > > > > **Following the definitions of confounding in the literature, we propose a way of measuring the strength of confounding among a set of variables.**
> > > > >
> > > > > However, as mentioned previously, exploring the concept of negative confounding could be an intriguing direction for future research. As noted in our earlier response, we will explain the importance of these properties before formally deriving them as the properties satisfied by our measures.
> > > > >
> > > > > We hope that this clarifies our perspective; we would be happy to respond further if possible.
> > > > >
> > > > > References:
> > > > >
> > > > > [14] Sander Greenland and Hal Morgenstern. Confounding in health research. Annual review of public health, 22(1):189–212, 2001.
> > > > >
> > > > > [40] Menglan Pang, Jay S Kaufman, and Robert W Platt. Studying noncollapsibility of the odds ratio with marginal structural and logistic regression models. Statistical methods in medical research, 25(5):1925–1937, 2016.
> > > > >
> > > > > [41] Judea Pearl. Causality. Cambridge university press, 2009.
> > > > >
> > > > > [51] Noah A Schuster, Jos WR Twisk, Gerben Ter Riet, Martijn W Heymans, and Judith JM Rijnhart. Noncollapsibility and its role in quantifying confounding bias in logistic regression. BMC medical research methodology, 21:1–9, 2021.

---

### Official Review · Reviewer_K6Rd · 2024-07-10

**Soundness:** 3
**Presentation:** 3
**Contribution:** 2
**Rating:** 6
**Confidence:** 4

**Summary:**

The paper addresses the challenge of identifying and quantifying (unobserved) confounding in causal inference. They propose a more comprehensive approach by relaxing the classic assumption of causal sufficiency and leveraging the sparse causal mechanism shifts assumption. The authors introduce methods to detect and measure confounding effects, distinguish observed and unobserved confounding, and evaluate the relative strengths of confounding among variable sets.  An empirical validation supports their theoretical analysis.

**Strengths:**

(S1) The paper provides a thorough study of confounding from several perspectives: detecting and measuring confounding between pairs of variables and among multiple variables; distinguishing between observed and unobserved confounding; assessing the relative strengths of confounding among different sets of observed variables. To my knowledge, this is the first study exploring all these aspects in a unified framework.

(S2) The paper is well written and easy to follow.

**Weaknesses:**

(W1) I found the paper to be lacking in experimental evaluations. It is not clear how hard (both statistically and computationally) it is to compute the measures of confounding that were proposed, especially the ones measuring confounding among multiple variables.

(W2) I am somehow skeptic about the practical relevance of the results presented in the paper (and the lack of experiments reinforces this point, see W1). In particular, people focused a lot on the marginal sensitivity model (and variations) as a measure of confounding strength because sensitivity analysis bounds can be easily derived from it. I am not sure the same would be possible under the proposed measure of confounding, and hence I am not sure how it would be useful in practice.


(W3) I think some relevant related works are missing. In particular, when interventional data is available (e.g. RCTs) [1] and [2] propose testable implications for detecting hidden confounding. Further, [3] and [4] propose to lower bound the strength of hidden confounding (as measured by the marginal sensitivity model).

[1] Falsification of Internal and External Validity in Observational Studies via Conditional Moment Restrictions. Hussain et al. AISTATS 2023.

[2] Benchmarking Observational Studies with Experimental Data under Right-Censoring. Demirel et al. AISTATS 2024.

[3] Hidden yet quantifiable: A lower bound for confounding strength using randomized trials. De Bartolomeis et al. AISTATS 2024.

[4] Detecting critical treatment effect bias in small subgroups. De Bartolomeis et al. UAI 2024

**Questions:**

(Q1) I got confused at lines 103-106: "While these methods use data from different contexts (an approach we also leverage in this work), they assume the absence of unobserved confounding variables; we instead focus on capturing both observed and unobserved confounding using data from multiple contexts.". [27] proposes a test for the presence of hidden confounding and hence allows for unobserved variables. Can the author briefly comment on how their measures compare with the approach proposed in [27]?

(Q2) I am confused slightly confused with Assumption 3.2 (line 169):  $$ P^c( X_i | \text{PA}_i) \neq P^{c'}(X_i | \text{PA}_i) $$
what does it mean for conditional distributions to be different?

(Q3) Can the authors comment with one motivating example of how their proposed confounding measure could be used in practice? (See W2)

[27] Rickard Karlsson and Jesse Krijthe. Detecting hidden confounding in observational data using multiple environments. Advances in Neural Information Processing Systems, 36, 2023.

**Limitations:**

Yes

---

> ### Author Rebuttal · Authors · 2024-08-07
>
> > I found the paper to be lacking in experimental evaluations. It is not clear how hard (both statistically and computationally) it is to compute the measures of confounding that were proposed, especially the ones measuring confounding among multiple variables.
>
> Due to space constraints, we included experiments in Appendix Section B (our paper required significant space for discussing the considered 3 settings and our formulations therein). We have now included additional experimental results in the uploaded rebuttal PDF. If accepted, we will use the additional page available in the final version to include the results in the main paper itself.
>
> As stated in the appendix, our experiments were conducted on a CPU and are straightforward to run. We have provided the code in the supplementary material, and it can be executed quickly.
>
> > I am somehow skeptic about the practical relevance of the results presented in the paper ... people focused a lot on the marginal sensitivity model ... I am not sure the same would be possible under the proposed measure of confounding, and hence I am not sure how it would be useful in practice.  I think some relevant related works are missing...
>
> In the uploaded rebuttal PDF, Figure 1 demonstrates two simple real-world examples where our methods can be applied (the purpose of these examples is only to show its practical value, and not intended to be comprehensive or complex). As explained in the previous response, we have included experiments in the Appendix and new experiments in the uploaded rebuttal PDF.
>
> We appreciate your suggestion regarding sensitivity analysis. We will incorporate the following points in the revised manuscript for completeness.
>
> 1. Our method aims to estimate the exact value of confounding rather than approximate it using bounds obtained via sensitivity analysis.
> 2. The difference between the total confounding and the conditional confounding by conditionining on observed confounding variables can be interpreted as the unobserved confounding estimate obtained via sensitivity analysis.
> 3. As discussed with reviewer n1WD, we can connect marginal sensitivity definition of confounding and the confounding based on directed information used in our paper. We will include this connection in the revised manuscript.
>
> We also thank you for providing the relevant papers. We will discuss them in the revised manuscript and include a discussion on sensitivity analysis as mentioned before for completeness.
>
> > Q1) I got confused at lines 103-106: "While these methods use data from different contexts... [27] proposes a test for the presence of hidden confounding and hence allows for unobserved variables... compare with the approach proposed in [27]?
>
>
> Apologies for the oversight. The study of hidden confounding detection in [27] primarily focuses on downstream causal effect estimation. In contrast, our work aims to provide a unified framework for studying and measuring both observed and unobserved confounding across different types of contextual information. This framework supports various downstream applications beyond causal effect identification, including assessing the relative strengths of confounding and measuring confounding between pairs and sets of variables. We will add these points to the related work in the revised manuscript. We also study downstream causal effect estimation tasks. As demonstrated in the results in the rebuttal PDF, confounding detection using our method contributes to improved causal effect estimation.
>
> > Q2) I am slightly confused with Assumption 3.2 (line 169): what does it mean for conditional distributions to be different?
>
> Consider a variable $X_i$ and its parents $PA_i$. If $\mathbb{P}(X_i=x_i|PA_i = pa_i)$ is different in two contexts/environments $c,c'$ for atleast one pair $(x_i, pa_i)$, we say that the causal mechanisms are different in two contexts $c,c'$. According to Assumption 3.2, such causal mechanism shifts are rare/sparse for a variable $X_i$.
>
> > (Q3) Can the authors comment with one motivating example of how their proposed confounding measure could be used in practice? (See W2)
>
> We have included two simple real-world examples in the uploaded rebuttal PDF (Figure 1) demonstrating the applicability of our methods. Additionally, since the benchmark datasets from the BNLearn repository come with known data-generating processes, we can generate different contexts by performing interventions and subsequently applying our method, similar to [35,37].
>
> References:
> [27] Rickard Karlsson and Jesse Krijthe. Detecting hidden confounding in observational data using multiple environments. Advances in Neural Information Processing Systems, 36, 2023.
> [35] Mameche, Sarah, Jilles Vreeken, and David Kaltenpoth. "Identifying Confounding from Causal Mechanism Shifts." International Conference on Artificial Intelligence and Statistics. PMLR, 2024.
> [37] Mooij, Joris M., Sara Magliacane, and Tom Claassen. "Joint causal inference from multiple contexts." Journal of machine learning research 21.99 (2020): 1-108.
>
>
> We hope our responses address your concerns. We will update the manusript accordingly. We are happy to discuss further if you have any additional questions.

---

> > ### Comment · Reviewer_K6Rd · 2024-08-07
> >
> > I thank the authors for their response, and I maintain my original score.

---

### Official Review · Reviewer_3wgM · 2024-07-10

**Soundness:** 3
**Presentation:** 3
**Contribution:** 3
**Rating:** 6
**Confidence:** 3

**Summary:**

The authors mainly introduce capturing both observed and unobserved confounding using data from multiple contexts. Leveraging experimental data proposes a comprehensive approach for detecting and measuring confounding effects from three different settings that don't need the parametric assumptions and relaxes the causal sufficiency assumption. The authors provide the measure for detecting confounding effects (relative strengths of confounding).
For each of the proposed measures, this article presents key properties and an algorithm for detecting and measuring confounding using data from multiple contexts.

**Strengths:**

The entire article primarily introduces the detection and quantification of confounding effects without making parametric or causal sufficiency assumptions. It presents a well-structured logical framework to discuss this concept.

**Weaknesses:**

I think overall the authors did interesting research, but my main concerns are listed below.

1. When the environment changes, for example, if $c$ changes to $c'$, will the original causal relationship change?

2. If the length of the environment node is only 1, will the method described in the article fail?

3. Why do settings 2 and 3 introduce mutual information to define confounding? Would there be any difficulties in using the KL divergence?

4. In line 104, the author said "an approach we also leverage in this work", but the author did not introduce the method in which article in detail.

5. In line 111, the article describes "we measure the effects of both observed and unobserved confounding". However, the subsequent sections do not appear to provide a direct method for measuring causal effects.


6. In fact, the confounding effect is not always a number between 0 and 1. Converting it to a number between 0 and 1 only measures its relative strength, but does not directly calculate its confounding effect.

7. In Setting 1: When there is confounding between two variables, CNF-1 is used to calculate it. If multiple variables have common confounding, CNF-1 is calculated for each variable with the others and then summed. Will this lead to the repeated accumulation of confounding?

8. The detect confounding method proposed in the article, but this measure is relatively rarely used in real life. In fact, it is unrealistic to only conduct simulation experiments. In this part of the experiments, the author did not specifically introduce the settings, so why were only the CNF-2 results output, and not the CNF-3 results? Is there a significant difference between these two results? For the section on "Measuring Conditional Confounding," the author should add experiments involving observed confounding, such as replacing the unobserved confounding with observed confounding in Experiment 1.

9. The authors repeat write “decision variables” in lines 149 or 150, and "two" in line 156.

10. The authors miswrite “$\mathcal{D^{C}}$” as “$\mathcal{D}$” in the third line of ALgorithm 1.

**Questions:**

See the above Weaknesses.

**Limitations:**

The authors provide a detailed discussion of the limitations and applicability of their method.

---

> ### Author Rebuttal · Authors · 2024-08-07
>
> > When the environment changes, for example, if c changes to c', will the original causal relationship change?
>
> It depends on the type of intervention. If context change is a result of soft intervention on a variable, the underlying causal relationships do not change. If the context change is a result of hard intervention, the causal relationships can change. A similar discussion can be found in Section 2 of [35]. In this paper, we consider both types of interventions on  variables as highlighted in Table 1.
>
> > If the length of the environment node is only 1, will the method described in the article fail?
>
> When there is only one context available, there is a fundamental limitation: confounding cannot be uniquely identified, as explained in lines 43-45. Rather than calling it a limitation, we state that our method is not applicable when there is only one environment.
>
> > Why do settings 2 and 3 introduce mutual information to define confounding? Would there be any difficulties ..?
>
> We build upon earlier works that utilize mutual information to measure confounding. According to our formulation, mutual information is among the most suitable choices for measuring the dependency between random variables induced by changes in the mechanism, as explained in lines 295-297. See [35] for a similar approach using mutual information for identifying confounding. In our experiments, we did not encounter any difficulties using KL divergence.
>
> > In line 104, the author said "an approach we also leverage in this work", but the author did not introduce the method in which article in detail.
>
> Apologies for the confusion. By "an approach," we refer to the idea of using different contexts to measure confounding. We will either modify or remove the text in brackets to clarify this point and avoid any confusion.
>
> > In line 111, the article describes "we measure the effects of both observed and unobserved confounding". However, the subsequent...
>
> We believe the reason for confusion is the word 'effect'. We aim to study and quantify the confounding **bias** instead of its effect on causal **effects** as a downstream application. Our goal in this paper is to provide a unified framework for confoudning. We will replace **effect** with **bias** to remove the ambiguity.
>
> We appreciate your interest in the results on causal effects. We conducted experiments to study the impact of the proposed confounding measures on reducing bias in estimated causal effects. Table 2 in the uploaded rebuttal PDF demonstrates that controlling for the variables detected as confounders by our method helps reduce the bias in estimated causal effects.
>
> > In fact, the confounding effect is not always a number between 0 and 1. Converting it to a number between 0 and 1...
>
> We provide a reason for studying relative strength in lines 57-62 of the introduction section. Also, it is trivial to evaluate the actual confounding effect that is not between 0 and 1 by ignoring the exponential transformation used in our definition of confounding measure.
>
> > In Setting 1: When there is confounding between two variables, CNF-1 is used to calculate it. If multiple variables have common confounding...
>
> Since this is a first effort to define a measure of confounding among a set of variables, our focus herein was on the stated settings. It is possible however to accumulate the confounding. It is easy to see that our current definition of measuring joint confounding  satisfies positivity and monotonicity properties. We believe that exploring other ways of defining joint confounding can be interesting directions of future work.
>
> > The detect confounding method proposed in the article, but this measure is relatively rarely ...
>
> We included two simple real-world applications of our method in the uploaded rebuttal PDF, as shown in Figure 1. (The purpose of these examples is only to show its practical value, and not intended to be comprehensive or complex.) For the conditional confounding experiments, we observe similar results for Settings 2 and 3; therefore, we report results for Setting 2 only. Additionally, our experiments on conditional confounding assume that the confounding variables are observed. We have provided additional experimental results in the uploaded rebuttal PDF, detailed in Tables 1 and 2, which demonstrate the usefulness of our methods.
>
> > The authors miswrite $D^c$ as $D$ in the third line of ALgorithm 1.
>
> As explained in line 245, we combine data from all contexts to evaluate conditional probability. We will update third line of Algorithm 1 to include $\mathcal{D} = \cup_{c}\{\mathcal{D}^c\}$ to make it clear.
>
> References:
> [35] Mameche, Sarah, Jilles Vreeken, and David Kaltenpoth. "Identifying Confounding from Causal Mechanism Shifts." International Conference on Artificial Intelligence and Statistics. PMLR, 2024.
>
> We hope our responses address your concerns. We will update the manusript accordingly. We are happy to discuss further if you have any additional questions.

---

> > ### Comment · Reviewer_3wgM · 2024-08-12
> >
> > Thanks for your response. The authors have addressed my main concerns, so I raise my score accordingly. I strongly recommend that the authors incorporate real-world applications into the main paper to enhance its overall completeness.

---

> > > ### Author Response · Authors · 2024-08-12
> > > **Thank you for reply**
> > >
> > > We thank you for your reply. We will include the real-world applications in the main paper for completeness.

---

### Official Review · Reviewer_n1WD · 2024-07-13

**Soundness:** 3
**Presentation:** 3
**Contribution:** 3
**Rating:** 5
**Confidence:** 4

**Summary:**

This paper presents methods that:
1. define a measure of confounding between sets of variables,
2. separate the effects of observed and unobserved confounders, and
3.  assess the relative strengths of confounding between sets of variables,

in three different settings. In each setting they assume that data from several different contexts is available, and that the changes in causal mechanisms between the different contexts are known.

**Strengths:**

This paper provides thorough theoretical justification for their methods. The analyses given for the latter two settings seem particularly novel to me.

**Weaknesses:**

The assumption "that the causal mechanism changes are known for each variable across different contexts" seems to be quite strong, but is not mentioned until relatively late in the paper (line 160).

The formal definition of a mechanism shift comes quite late in the paper. It would be nice to have intuitive explanations in the abstract and introduction. I also wonder if the formal notation could start with the notion of multiple environments baked in. As it is written now, it is unclear what the mechanism shift is relative to. Is it a shift between an environmental distribution and the interventional distribution, or is it only necessary to know the relative shift between environments?

The relationship between definitions 4.2 and 4.3 and ignorability and conditional ignorability as independence and conditional independence assumptions needs to be added to the related works. There is also perhaps a connection to [Scalable Sensitivity and Uncertainty Analyses for Causal-Effect Estimates of Continuous-Valued Interventions](https://arxiv.org/pdf/2204.10022) that could be discussed. Notably, the relationship between the likelihood ratios and the KL divergence between the interventional and observational distribution mentioned in Appendix A.3.1.

While the theoretical results are thorough, the empirical results are very limited. A major concern with hidden confounding is that it can arbitrarily bias estimated treatment effects. The magnitude of the bias should be straightforward to calculate in your synthetic experiments. It might be more demonstrative to show how the proposed measures correlate to induced biases.

## Points of confusion

**Definition 3.1**
- Looks like $X$ is being redefined here as to include any variable in $V$ rather than the subset that excludes $Z$. Mildly confusing.
- line 131: $P(X_i \mid \mathrm{PA}_i)$ is not defined in definition 3.1

**Assumption 3.1**
- talks about changes in causal mechanisms, but this concept hasn't been formalized yet which makes this assumption hard to understand.

**Line 147:** what is a context node? what is an extended causal graph? Might help to define context specific distributions before rather than after.

**Line 156:** two two

**Line 156:** $P^c$ notation could be introduced earlier to formalize a mechanism shift.

**Assumption 3.3:** $dPC$ is missing brackets

**Line 182.** shifts -> shift

**Line 183:** "extent of hidden confounding." word choice is a bit vague. maybe strength, or the effect of the confounder on each variable, or how much it may bias some estimand.

**Definition 4.1:** The notation is a little awkward with the condition on $P(X_j)$ within $D_{\mathrm{KL}}$.

**Theorem 4.2:** strongly -> more strongly

**Definition 4.6:** missing definition of mutual information.

**Questions:**

**line 128:** unclear notation, what information is $P_x(V), X \subseteq V$ adding here? If $P_*$ is a set, why not write the set that defines it instead?
**line 128:** (i.e., $X = \emptyset$). why isn't $X$ bold?

**Line 154:** "Let $C_{S \wedge \neg R}$ be the set of contexts in which we observe mechanism changes for the set of variables $X_S$ but not for the variables $X_R$." Changes relative to what? The interventional distribution? Just between two contexts? Is the shift the same across each of the contexts in this set?

**Line 158:** This does not look like a p-value. This looks like an indicator taking a value of true or false, not a probability. Is there a missing $P$ outside of the brackets? If so, what is the relevant random variable? $c$? If $c$, confusing to have it lower-case while a potential value $X_i$ is upper case. Same questions for **Assumption 3.2**.

**Theorem 4.1:** missing comma? $\\{X_iX_j\\}$ -> $\\{X_i, X_j\\}$

**Line 286:** Are the $\epsilon$ normally distributed? or is zero mean and any distribution ok?

**Definition 4.7:** mutual information is defined between random variables. Are these random variables with respect to contexts? If so lower case $c$ is a bit confusing.

**Limitations:**

The authors adequately discuss the limitations.

---

> ### Author Rebuttal · Authors · 2024-08-07
>
> > The assumption "that the causal mechanism changes are known for each variable across different contexts" ..
>
> To measure confounding between **all pairs** of nodes in a causal graph, we need to know the mechanism changes for **each variable** across contexts. However, if the number of nodes among which confounding is to be identified is small, **we only need to know that the causal mechanisms are changed for that small set of nodes, but not for all nodes**. This assumption has been made in [35, 37]. We will edit line 160 as follows to make it clear that the assumption is not too strong.
>
> "*Hence, we focus on detecting and measuring confounding among a set of variables, assuming that the causal mechanism shifts are observed among that subset of variables.*"
>
> >The formal definition of a mechanism shift..?
>
> We thank you for the suggestions. We will ensure that the intuitive explanations of mechanism shifts appear early in the paper. In particular, we will move lines 131-133 and lines 163-169 to introduction line 48 to introduce causal mechanisms and how the shift in causal mechanisms are useful in detecting confounding. Also, it is sufficient to know relative shift between environments, where each environment is a result of either a soft or hard intervention to a set of variables.
>
> >The relationship between definitions 4.2 and 4.3 and ignorability....
>
> This is an interesting insight. We thank you for bringing up this point. The above paper presents an intriguing way of defining confounding as the difference between nominal and complete propensity scores in the potential outcomes framework. This is inherently connected to the ignorability assumption. Since directed information also relies on the KL-divergence between conditional and interventional distributions, we can leverage this definition to define relative confounding strength between sets of variables. We will include all of these points in the revised manuscript and discuss them in the related work section.
>
> > While the theoretical results are thorough...
>
> Our primary focus in this work was on effectively identifying confounding variables, which is non-trivial by itself. Once we effectively identify observed confounding variables, we can control for them to reduce bias in estimated causal effects. We appreciate your interest in the results on bias in estimated causal effects. We conducted experiments to study the impact of the proposed confounding measures on reducing bias. Table 2 in the uploaded rebuttal PDF demonstrates that controlling for the variables detected as confounders by our method helps reduce the bias in estimated causal effects.
>
> > Definition 3.1 Looks like X is being redefined here ...
>
> **Definition 3.1:** We undestand the concern. We will use a different variable in Defintion 3.1, say $\mathbf{W}$, in place of $\mathbf{X}$. With this change, none of the other contents gets affected.
>
> **Line 131:** We will edit line 131 as follows.
>
> *"For a node $X_i$, $\mathbb{P}(X_i|\mathbf{PA}_i)$ is called the *causal mechanism* of $X_i$."*
>
> > Assumption 3.1 talks about changes in causal mechanisms, but this concept hasn't been formalized...
>
> Assumption 3.1 discusses independent causal mechanisms but not changes in causal mechanisms. We introduce the concept of causal mechanism shifts in lines 145-146 and later in Assumption 3.2, we discuss sparse causal mechanism shifts.
>
> > Line 147: what is a context node?...
>
> Thank you for your suggestions. A context node can be viewed as an exogenous parent to the set of nodes on which an intervention is performed. Our analysis does not depend on this extended causal graph. However, we will define context specific distributions as you suggested to make this clear.
>
> > Definition 4.1: The notation is a little awkward with the condition on $\mathbb{P}(X_{j})$ within $D_{KL}$.
>
> We followed [60] to use this particular notation of conditioning on $\mathbb{P}(X_{j})$ within $D_{KL}$. We believe this is to differentiate between conditional and interventional probabilities. Since it is clear from the context, we will exclude conditioning on $\mathbb{P}(X_{j})$ in the revised manuscript.
>
> > Line 128: unclear notation, what information is $P_{x}(V), X\subseteq V$ adding here?...
>
> We use $P_x (V), X \subseteq V$ and $P_*$ to formally define causal Bayesian networks. They do not have any relation to our method. $X$ should be bold here. We will fix this typo.
>
> > Line 154: "Let $C_{S\wedge \neg R}$ be the set of contexts in which we observe mechanism changes for the set of variables $X_S$ but not for the variables $X_R$." Changes relative to what?...
>
> In your question, if $i\in S$, we have $P^c (X_i|\mathbf{PA}_i) \neq P^{c'}(X_i|\mathbf{PA}_i)$. If $i \in R$, the inequality becomes equality.  The shifts do not have to be the same across all contexts. Contexts are a result of either soft or hard intervention on the nodes. Hence, the changes between contexts are relative to interventional distributions.
>
> >Line 158: This does not look like a p-value. This looks like an indicator taking a value of true or false, not a probability... Same questions for Assumption 3.2.
>
>  Following [35], we will edit line 158 as follows:
>
> *"For example, the $p\text{-value}(\mathbb{P}^c(X_i|\mathbf{PA}_i^o)\neq\mathbb{P}^{c'}(X_i|\mathbf{PA}_i^o))$ where $\mathbf{PA}_i^o$ is the set of..."*
>
> The above p-value tests whether the causal mechanisms of $X_i$ are different in two different contexts $c,c'$. We will remove $p$ from line 158 to avoid confusion. In assumption 3.2, similar to [35], we use $p$ to indicate the probability of two distributions being different.
>
> >Line 286: Are the $\epsilon$ normally distributed? or is zero mean and any distribution ok?
>
> The only restriction on $\epsilon$ is that it has zero mean with no other restriction on the underlying probability distribution.
>
> > Definition 4.7: Lower case $c$ is a bit confusing.
> We will replace $E^c_i$ to $E^C_i$ to avoid confusion in the revised manuscript.

---

> ### Comment · Reviewer_n1WD · 2024-08-12
>
> Thank you for replying to my review. After reading your responses, I have decided to raise my score to a 5. I still think this paper needs significant revision. I choose to trust that the changes promised to all reviewers will be made, and leave the final decision to the AC.

---

> > ### Author Response · Authors · 2024-08-13
> >
> > We sincerely appreciate your time, insightful comments, and positive response. We will update the manuscript by incorporating all the suggestions from the reviewers. We have provided detailed information on our planned improvements in a common response to all reviewers above. Please see our response here: https://openreview.net/forum?id=SvmJJJS0q1&noteId=ukyx7Qc4m9

---

### Author Rebuttal · Authors · 2024-08-07

## Common response to all reviewers
We thank all reviewers for their thoughtful feedback. We are pleased to see the following encouraging comments from the reviewers.

1. The problem addressed in this paper is of significant importance (e4Fq).
2. The ideas presented are both original (e4Fq) and novel (n1WD).
3. The theoretical justification for our methods is thorough (n1WD).
4. We are the first to study various aspects of confounding (K6Rd) within a unified framework (K6Rd, 3wgM).
5. The paper is well-written and well-presented (K6Rd, e4Fq).

As can be seen from the reviews, most of the reviewers' concerns are about clarifications, which we have addressed below. We have also uploaded a PDF with additional results as suggested by the reviewers. Our paper required significant space for discussing the considered 3 settings and our formulations therein; for purposes of clarity, the experimental results were moved to Appendix. If accepted, we will use the additional page available in the final version to include the results in the main paper itself. The additional results, included in the rebuttal, further demonstrate the usefulness of our methods. We will update the manuscript as per the suggestions, and will release our code publicly on acceptance.

---

### Author Response · Authors · 2024-08-13
**A Note to All Reviewers and AC**

We sincerely appreciate all reviewers for their time, insightful comments, and positive feedback.  Reviewers' suggestions have significantly contributed to improving the presentation of our work. The majority of the suggestions from reviewers pertain to clarifications and editorial changes, which we addressed to the best of our knowledge (as also acknowledged by most reviewers).

The reviewers' suggestions and comments are resolvable and feasible to be included within the allowed 10 pages (we will use the additional page granted if the paper gets accepted). We have prepared an updated manuscript incorporating the following changes:

1. We have moved the experiments from the appendix to the main paper and added the new experiments shown in the rebuttal PDF to the main paper.
2. The two real-world examples presented in the rebuttal PDF are now included in the introduction.
3. The relationship to sensitivity analysis, as suggested by reviewers K6Rd and n1WD, has been added to the related work and methodology sections of the revised manuscript.
4. All typos and other editorial comments have been addressed in the final manuscript.

---

### Decision · Program_Chairs · 2024-09-25

**Decision:**

Accept (poster)

**Comment:**

This paper introduces a novel approach for detecting and quantifying confounding effects by leveraging known causal mechanism shifts in observed variables, without relying on parametric assumptions or causal sufficiency. The idea is recognized as novel and original (n1WD, e4Fq), the paper is well-written (K6Rd, e4Fq), and the work is presented within a unified framework (K6Rd, 3wgM). However, some concerns remain regarding the clarity of certain descriptions (n1WD), the depth of experimental evaluations, and the demonstration of real-world applicability (3wgM, K6Rd, e4Fq). Additionally, the importance of the proposed measure for confoundedness requires further consideration (e4Fq). Nonetheless, the authors have adequately addressed the key issues in their responses, leading to the decision to accept the paper.